# Canine urothelial cell model to study intracellular bacterial community development by uropathogenic *Escherichia coli*

Jessica M. Gilbertie[1,2], Breanna J. Sheahan[2,3], Shelly L. Vaden[3], Megan E. Jacob[1,2]*

1 Department of Population Health and Pathobiology, College of Veterinary Medicine, North Carolina State University, Raleigh, North Carolina, United States of America, 2 Comparative Medicine Institute, North Carolina State University, Raleigh, NC, United States of America, 3 Department of Clinical Sciences, College of Veterinary Medicine, North Carolina State University, Raleigh, North Carolina United States of America

* mejacob@ncsu.edu

**Data Availability Statement:** All relevant data are within the manuscript and its supplementary figures.

## Abstract

Urinary tract infections (UTIs) are among the most common bacterial infections of both dogs and humans, with most caused by uropathogenic *Escherichia coli* (UPEC). Recurrent UPEC infections are a major concern in the treatment and management of UTIs in both species. In humans, the ability of UPECs to form intracellular bacterial communities (IBCs) within urothelial cells has been implicated in recurrent UTIs. However, the role of IBCs has not been explored in the pathogenesis of canine recurrent UTIs. In this study, we identified IBCs in both urine and bladder tissue from dogs with UPEC associated UTIs. In addition, we showed that UPECs derived from canine UTIs form IBCs within primary canine urothelial cells. As in human UTIs, formation of IBCs by canine UPECs correlated with the presence of the *fimH* gene as those isolates lacking the *fimH* gene formed fewer IBCs in canine urothelial cells then those harboring the *fimH* gene. Additionally, UPEC strains from clinical cases classified as recurrent UTIs had higher rates of IBC formation than UPEC strains from non-recurrent UTIs. These IBCs were tolerant to treatment with enrofloxacin, cefpodoxime and doxycycline at 150, 50 and 50 μg/mL respectively, which are representative of the concentrations achieved in canine urine after standard dosing. This is consistent with the clinical perspective that current UTIs are a common condition of dogs and are difficult to manage through antimicrobial treatment. Additionally, the dog could prove to be a powerful model of IBC formation as they are natural models of UPEC-causing UTIs and have similar pathophysiology of IBC formation.

## Introduction

Urinary tract infections (UTI) are among the most common bacterial infections of dogs accounting for 14% of medical cases. Uropathogenic *Escherichia coli* (UPEC) are the most common etiological agents in both human and canine UTIs [1,2]. Recurrent infections are a major concern in the treatment and management of UTIs of both species [3]. Previous reports

**Funding:** The author(s) received no specific funding for this work.

**Competing interests:** The authors have declared that no competing interests exist.

show that 20% of dogs with UTIs will develop a recurrent infection with the same uropathogen, despite antimicrobial therapy, within 6-months of the original infection [4]. In humans, recurrent UTIs are associated with the ability of UPEC to form intracellular bacterial communities (IBCs) that are tolerant to antimicrobial therapy [5–8]; however, the role of IBCs in canine UTIs has not been demonstrated.

There has been an increasing interest in the relationship between UPEC biofilm potential and infection persistence in recurrent UTIs; however, a previous study from our group was unable to identify associations between traditional biofilm formation *in vitro* and clinical recurrence of UTI in dogs [9]. The traditional definition of a biofilm is a community of bacteria within a polymeric matrix that is attached to an abiotic or biotic surface [10]. However, recent advancements in biofilm research suggest that, in the urinary tract, IBCs are considered non-surface attached, intracellular biofilms [11].

In human isolates, the ability of UPEC to adhere to and invade the urothelium to form IBCs is facilitated by the type 1 pilus-associated adhesin FimH [12]. The majority of UPECs encode the *fimH* gene, a part of the *fim* operon, which is transcribed to the FimH adhesion protein [13]. FimH binds to mannose-containing glycoprotein receptors on host cell membranes, which in the urinary tract, are the mannosylated residues on uroplakin proteins that line the apical surface of urothelial cells in the bladder [14]. UPEC utilize binding of FimH to the uroplakin receptor complex to enable adhesion to the urothelial surface [12]. Subsequently, the binding of FimH to uroplakin triggers host cell signaling promoting bacterial internalization [15]. Thereafter, UPEC grow and divide intracellularly forming small clusters of bacteria termed IBCs [16]. Once an IBC develops, UPEC are protected, able to tolerate the -cidal activity of many antimicrobials [6,17].

Dogs are an ideal translational model for urinary tract infection research as their size allows for repeated access to large volumes of urine [18] and biopsies of bladder tissue can be easily accomplished via uroscopy. In addition, dogs are commonly used as models of human disease and are great pharmacokinetic models for novel therapeutics [19–21]. Also, dogs have similar UTI epidemiology and have urinary biology comparable to humans [2,4,22,23]. For UPEC specific UTIs, dogs are infected by similar sequence types (STs), have similar rates of multi-drug resistant strains, and canine *E. coli* isolates have shown similar virulence traits and biofilm phenotypes as human isolates [24–26]. However, it is currently unknown if canine UTIs have a similar pathogenesis to human UTIs. More specifically if IBCs are part of UTI pathogenesis in canines.

Therefore, the goal of this study was to determine if dogs demonstrate similar UPEC associated UTI pathogenesis predominated by IBC formation as observed in humans and murine models [27]. Through retrospective data analysis, we demonstrated that dogs are clinically afflicted by similar uropathogens with similar resistance rates and recurrent infection frequencies as human UTIs. In clinical specimens, we demonstrated that UPEC-causing UTIs were associated with IBCs on both urine cytologies and bladder biopsies in symptomatic dogs; additionally, we validated an *in vitro* IBC model using primary canine urothelial cells. Using this *in vitro* IBC model, we validated the role of *fimH* in IBC formation in urothelial cells, assessed IBCs tolerance to treatment with several classes of clinically relevant antimicrobials, and compared IBCs *in vitro* from UPEC isolated from recurrent UTI cases compared to UPEC isolated from nonrecurrent UTI cases.

## Results

### Canine UTIs exhibit similar uropathogen distribution, antimicrobial resistance and recurrent infection frequencies as human UTIs

To compare the epidemiology of canine and human UTIs we performed a retrospective analysis of canine uropathogens collected at a veterinary teaching hospital over a 5-year period

(2015–2019). During that time, 797 canine urine isolates were obtained, of which 87.3% were monospecies cultures and 12.7% were polymicrobial. The most common UTI associated pathogen was *E. coli* (39.1% of positive cultures), followed by *Enterococcus* spp. (15.7%), *Streptococcus* spp. (10.7%), *Staphylococcus pseudintermedius* (8%), *Klebsiella pneumoniae* (5.6%), and *Proteus mirabilis* (5.6%) (Fig 1A). In humans, *E. coli* is the dominant uropathogen as well; however, occurrence is higher with reported rates from 65–75% [28,29]. The second most common UTI associated pathogen of humans varies by study and are similar to canine UTI associated pathogens including *Staphylococcus saprophyticus*, *K. pneumoniae*, *P. mirabilis* and *Enterococcus* spp. [30]. Of the polymicrobial infections in our analysis, the most commonly isolated combination was *E. coli* and *Enterococcus* spp. (typically *faecium* and *faecalis*) (24.5%). The *E. coli-Enterococcus* spp. combination is also seen in approximately 36% of human polymicrobial UTIs [31]. The sex distribution of affected dogs was 60.9% female (female intact and female spayed) and 39.1% male (male intact and male castrated) and the median age at presentation was 5.6 years (middle aged adult). Similarly, in human medicine a higher rate of UTIs is observed in women and the highest UTIs rates are seen patients from 18–65 years of age [32].

Of the *E. coli* isolates obtained from either mono- or polymicrobial canine UTIs, 45% were defined as multidrug resistant (MDR; resistant to ≥ 3 antimicrobial classes) (Fig 1B) and 33% demonstrated resistance to beta-lactam antimicrobials based on the presence of *blaCMY* and/or *blaTEM* genes (Fig 1C). In the human literature, 49–68% of UPEC are classified as MDR and 24–40% were extended spectrum beta-lactamase (ESBL) producers [33–35].

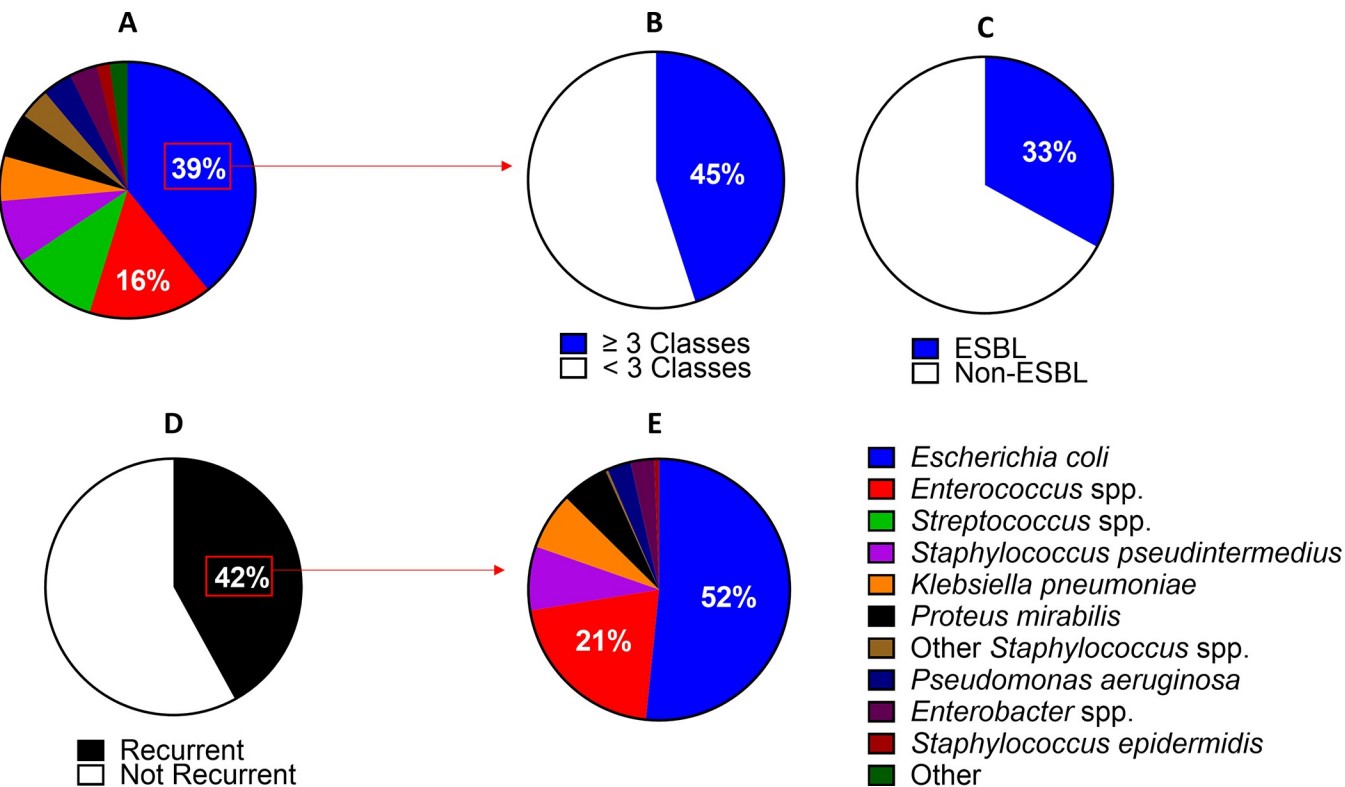

**Fig 1. Frequency of various uropathogens in canine urinary tract infections.** (A) Pathogen occurrence of all positive urine cultures (n = 797). Corresponding medical records were reviewed to determine if the canine patient was experiencing a recurrent UTI defined by more than one positive culture of the same uropathogen within 6 months of one another. (B) Occurrence of multidrug resistance in the *E. coli* population (n = 312) defined by resistance to greater than or equal to three antimicrobial drug classes. (C) Antimicrobial susceptibility profiles and gene presence of common ESBL genes, which are standard characterization parameters within the Clinical Microbiology Laboratory for any *E. coli* were reviewed. Occurrence of at least 1 positive ESBL gene in the *E. coli* population only. (D) Recurrence rates observed over the 5-year period of the retrospective (E) Pathogen presence or urine cultures from a recurrent infection.

Canine patients diagnosed with a recurrent UTI, defined as more than 1 positive urine culture within 6 months of the first urine culture and with the same uropathogen despite antimicrobial therapy, occurred in 42% of cases (Fig 1D). Comparable recurrence rates of 27–53% are observed in human UTIs [28,32]. In parallel to human studies, the most common bacterial pathogens causing recurrent UTIs in our retrospective analysis were *E. coli* (51.6%) and *Enterococcus* spp. (20.8%) (Fig 1E) [36].

## IBCs and filamentous bacteria are observed on urine cytologies and bladder biopsies from dogs diagnosed with UPEC associated UTIs

During human and murine-model UPEC pathogenesis, the bacteria bind to and invade into the cells of the urothelium where they replicate and form IBCs [11,16,37]. After IBCs mature, they will disperse out of the cell to infect neighboring facet cells [38]. When the bacteria emerge from the urothelial cell, they adopt a filamentous phenotype. Both IBCs and filamentous bacteria have been identified in the urine of humans with UTIs [27]. To determine if dogs with naturally occurring UTIs exhibit the presence of IBCs and filamentous bacteria, we evaluated urine cytologies (n = 30) and bladder biopsies (n = 10) from dogs with UPEC associated UTIs. Immunofluorescence was used to identify IBCs and filamentous bacteria (Fig 2). In our analysis of urine cytologies by immunofluorescence, we found UPEC IBCs in 60% of the cytologies evaluated (Fig 2A; 18 of 30) and observed filamentous bacteria in 77% of the cytologies

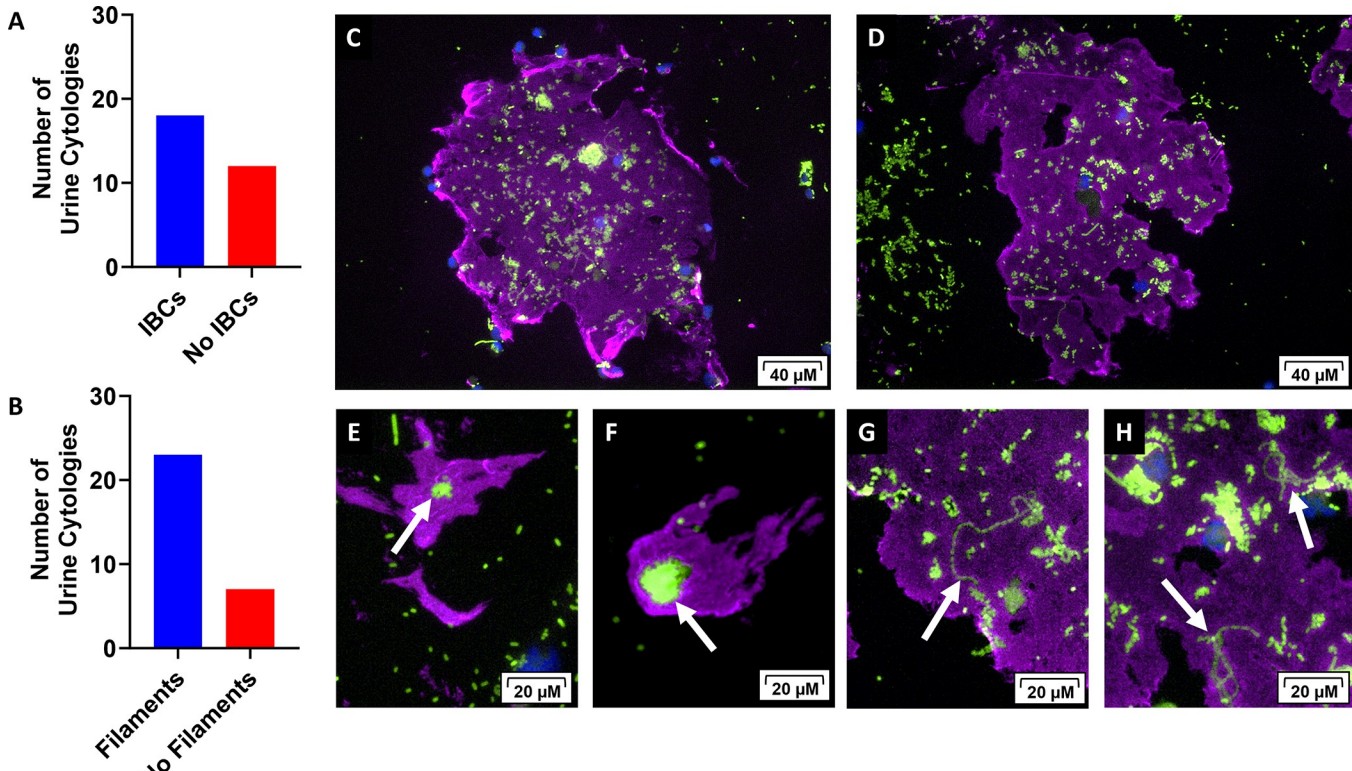

**Fig 2. Canine urine cytologies show evidence of IBCs and filamentous bacteria.** Urine cytologies from dogs with clinical symptoms of a UTI and a positive *E. coli* culture (n = 30) were stained with an anti-E. coli FITC (green) antibody, an anti-uroplakin IIIB antibody with an AlexaFluor 647 secondary antibody (purple) and DAPI (blue) to stain nuclei. (A) The number of urine cytologies evaluated that had observable IBCs within shed urothelial cells. (B) The number of urine cytologies evaluated that had observable filamentous bacteria. (C, D) Lower magnification of large urothelial cell sheets containing both IBCs and filamentous bacteria. (E, F) Higher magnification with white arrows pointing to discrete IBCS within single shed urothelial cells. (G, H) Higher magnification with white arrows pointing to filamentous bacteria emerging from the urothelium.

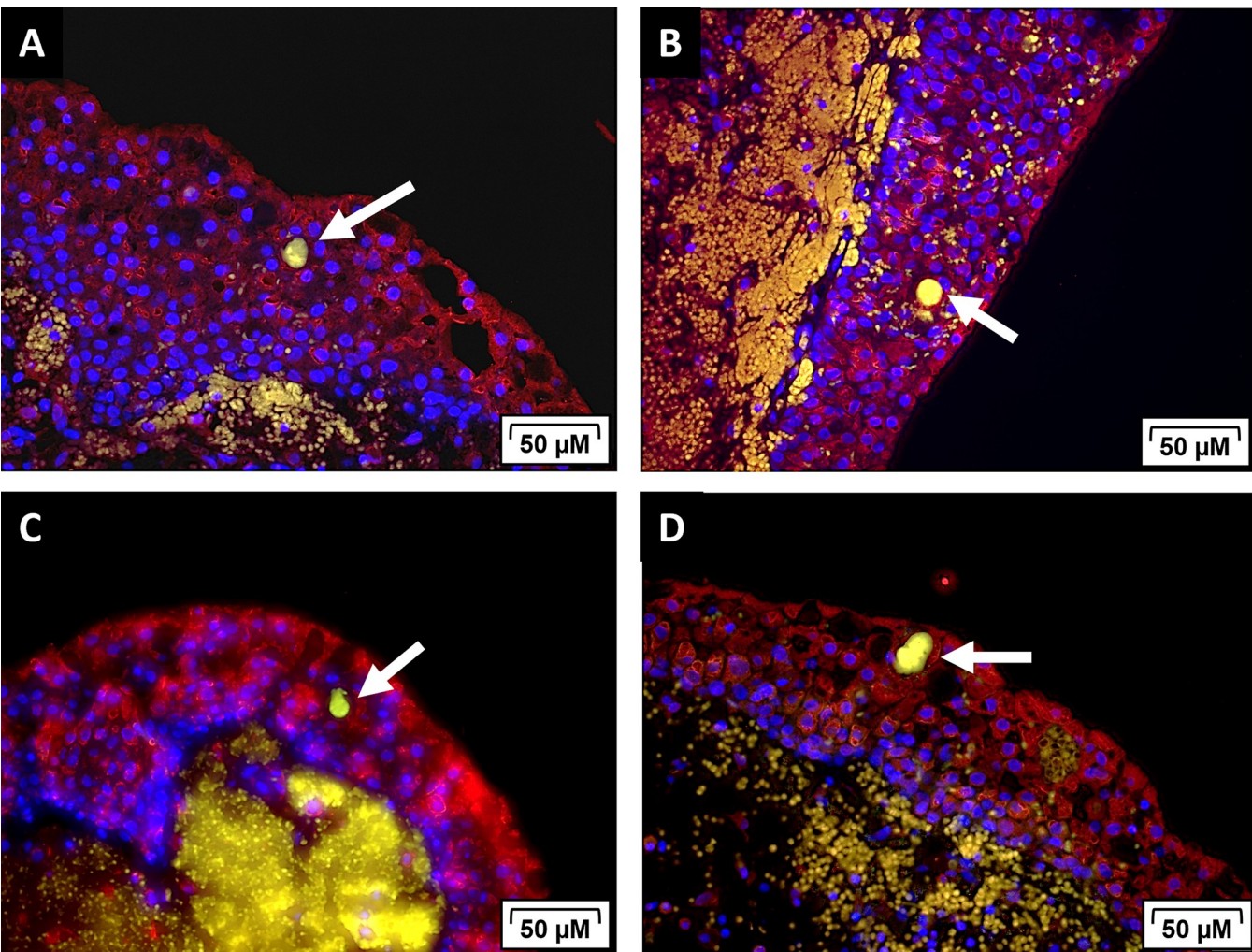

**Fig 3. IBCs are observed within the urothelium of dogs with UPEC-causing UTIs.** (A-D) Representative images of IBCs noted within the bladder epithelium of four different canine patients. Bladder biopsies were only evaluated from dogs with clinical signs of a UTI and a positive UPEC culture. Tissue samples were stained with the following antibodies: E-cadherin with an AlexaFluor 555 secondary (red), E. coli-FITC (green) and DAPI to stain nuclei (blue).

evaluated (Fig 2B; 23 of 30). Specifically, IBCs (anti-*E. coli*, green) were appreciated within sheets of urothelial cells (anti-uroplakin III, purple) (Fig 2C and 2D) and individual urothelial cells (Fig 2E and 2F). We also identified filamentous bacteria dispersing from the urothelium-associated IBCs (Fig 2G and 2H). Upon evaluation of the bladder biopsies, we observed IBCs (anti-*E. coli*, green) within the urothelium (anti-Ecadherin, red) in 4 out of the 10 dogs evaluated (Fig 3). The morphology of the IBCs observed in these clinical canine samples reflect those seen in humans and mice [21].

## Canine urothelial cell cultures support the formation of IBCs by clinical UPEC isolates

Previous research has shown that an immortalized human bladder epithelial cell line (PD07i) supports the growth of IBCs by UPEC *in vitro* [39]. Our aim was to develop an *in vitro* canine UTI model using canine primary urothelial cells and clinical canine UPEC strains. Urothelial cells harvested and cultured from canine bladders (n = 3) (S1 Fig) [40] were infected with

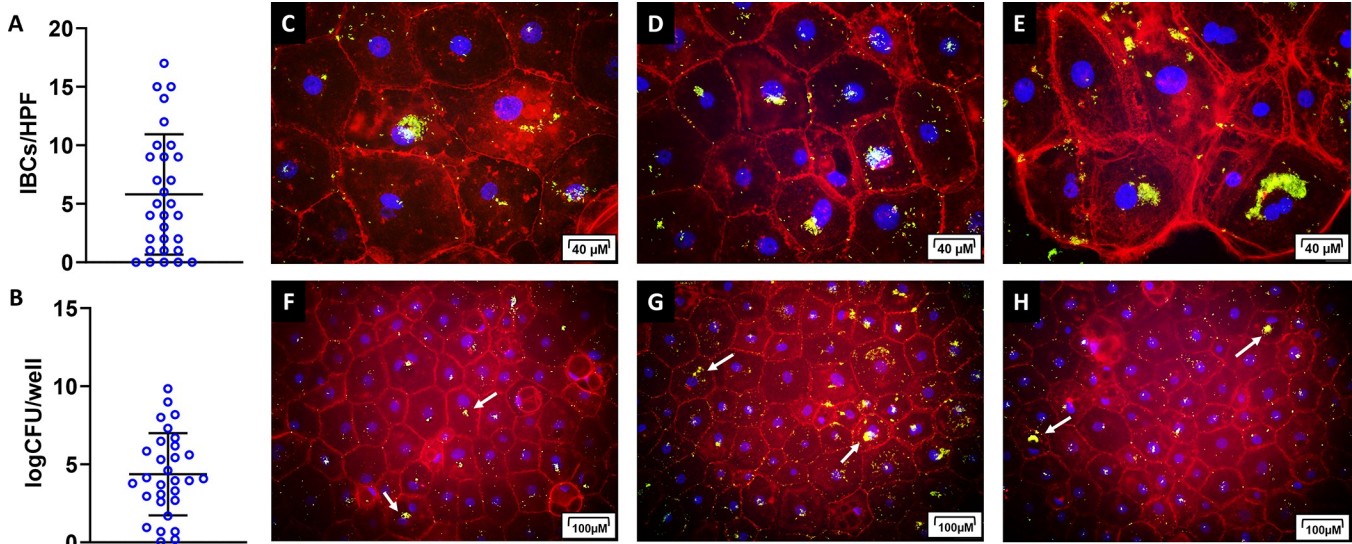

**Fig 4. Canine primary urothelial cells support IBC formation by canine associated UPECs.** Urothelial cells isolated from canine bladders (n = 3) were infected with clinical UPEC isolates from clinical canine UTIs (n = 30) in triplicate for each canine donor. Infected cells wer stained with an anti-E. coli FITC antibody (green), phallodin (red) to stain actin and DAPI (blue) to stain nuclei. (A) The number of IBC per high powered field (HPF) were counted for each donor/UPEC combination. The mean and standard deviation for the number of IBCs is shown by the black lines with the average of each UPEC isolate counted in each donor and in each replicate shown by the open blue circles. (B) The bacterial load per well was measured by lysing the cells in a second culture that was unfixed to release the intracellular bacteria. Data is represented as a log CFU per well. The mean and standard deviation for logCFU/mL is shown by the black lines with the average logCFU/mL for each UPEC isolate counted in each donor and in each replicate shown by the open blue circles. (C-E) High-powered representative images (20x) of IBCs observed in infected canine urothelial cells with various canine origin UPEC stains. (F-H) Low-powered representative images (10x).

UPEC strains originally isolated from symptomatic canines with confirmed UTIs (n = 30). At 24 hours post infection, we observed several of the UPEC strains forming IBCs within our canine urothelial cell model with an average of 5.8±5.15 IBCs observed per high powered field (HPF) (Fig 4A) and an average of 4.3±2.63 $\log_{10}$CFU per well (Fig 4B). Five of the 30 (16.7%) UPEC isolates evaluated had no observable IBCs, correlating with <1 $\log_{10}$CFU/mL (Fig 4A and 4B). The average size of canine IBCs was 29.2±16.1μm ranging from 10 μm to 80 μm (S2 Fig). The IBCs of the canine clinical UPEC strains grown in canine primary urothelial cells exhibited similar morphology to the IBCs of human UPEC strains grown in an immortalized bladder cell line (Fig 4C–4H) [39,41].

## IBC formation in canine urothelial cell culture is dependent upon *fimH*

Previous research in humans has shown that IBC formation is dependent upon the presence and expression of type 1 pilus or FimH in UPEC [12,42]. Therefore, we infected canine urothelial cells from three donors (n = 3) with clinical canine UPEC strains that were *fimH* positive (n = 9) or negative (n = 9) to determine the association with IBC formation. We found that clinical UPEC isolates containing the *fimH* gene formed significantly more IBCs in canine urothelial cells (7.7 IBCs/HPF) as compared to isolates lacking the *fimH* gene (0.5 IBCs/HPF) (Fig 5A, p<0.0001). The presence of *fimH* also affected the *E. coli concentration* within urothelial infections, with *fimH* positive isolates establishing a higher CFU/well (5.1 $\log_{10}$CFU/well) as compared to *fimH* negative isolates (0.8 $\log_{10}$CFU/well) (Fig 5B, p<0.0002). Representative images of urothelial infections from a *fimH*- UPEC infection (Fig 5C) and *fimH*+ UPEC infection (Fig 5D) further suggest that canine origin UPEC stains lacking *fimH* are unable to establish robust IBCs in canine urothelial cells which correlates with results seen in the murine model [12,43,44].

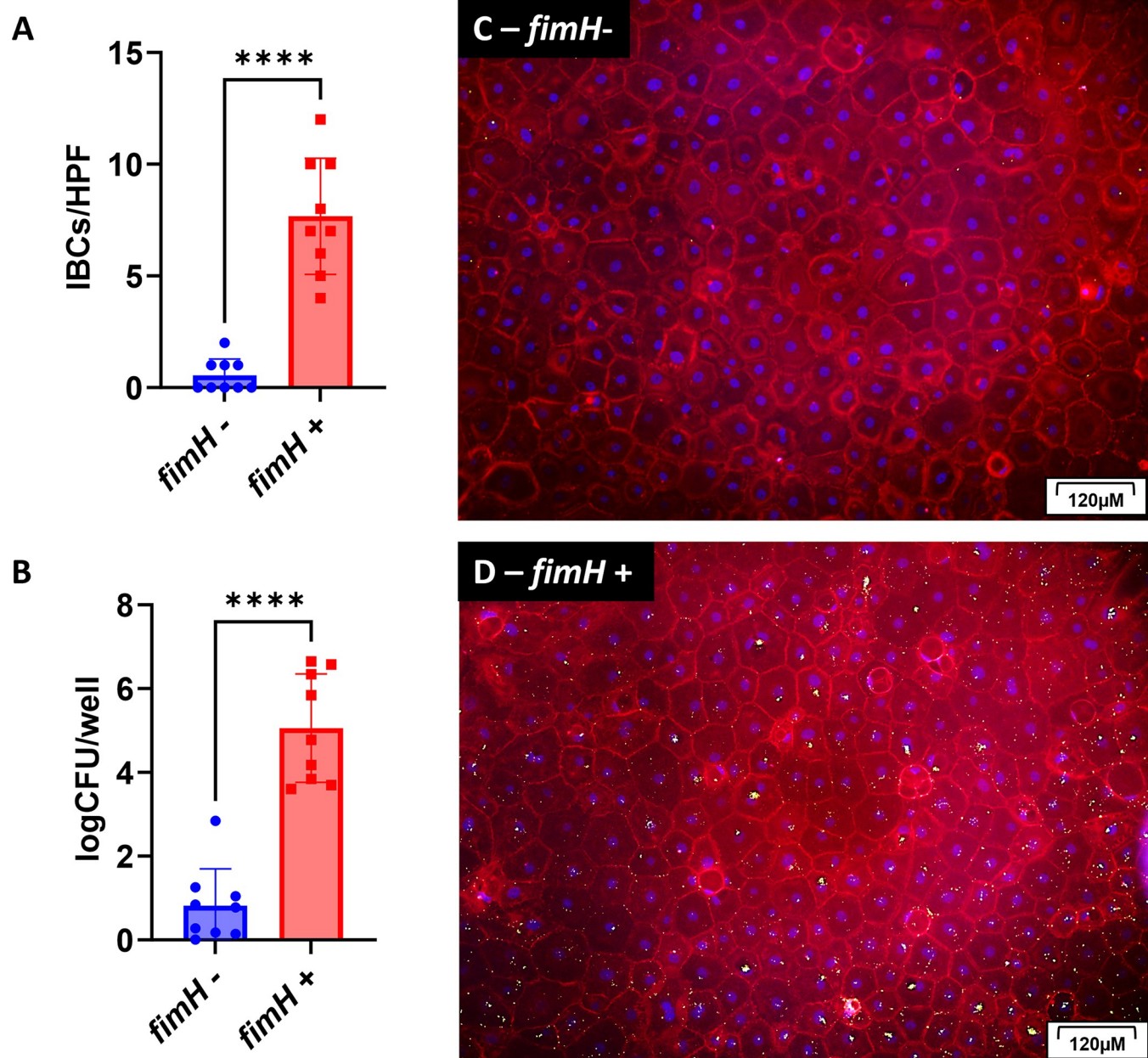

**Fig 5. Canine UPEC isolates with with the *fim*H gene form more robust IBCs compared to isolates lacking *fim*H.** Urothelial cells from canine donors (n = 3) were isolated, cultured, infected and evaluated as in Fig 4. Specifically, canine urothelial cells were infected with canine UPEC isolates lacking the *fimH* gene (n = 9) or those that harbored the *fimH* gene (n = 9). (A) The number of IBCs per high powered field (HPF) were counted for each donor/UPEC combination. The graph represents the average number of IBCs for each UPEC isolate counted in each donor and in each replicate. (B) The bacterial concentration per well was measured by lysing the cells in a second culture that was unfixed to release the intracellular bacteria. Data is represented as a log CFU/well. (C) Representative images of a urothelial cell infection by a *fimH-* UPEC. (D) Representative images of a urothelial cell infection by a *fimH+* UPEC. Bars are means and standard deviations and significant differences (p<0.05) were determined by an unpaired, nonparametric t-test (for IBCs/HPF) and parametric t-test (for log CFU/mL). Statistically significant differences are denoted with asterisks as follows: *p<0.05, **p<0.01, ***p<0.001, and ****p<0.0001.

## Canine urothelial cell IBC formation is more robust in UPEC isolates derived from cases of recurrent UTIs

To date, the genotypic or phenotypic relationship between UPEC strains and recurrent UTIs is unknown. Due to the variability in clinical UPEC isolates ability to form IBCs, we hypothesized that UPEC strains from recurrent UTI cases would form more IBCs *in vitro*. Above we showed that IBC formation was reliant upon the presence of *fimH*. Interestingly, UPEC isolates from recurrent UTIs were positive more frequently for *fimH* (97.5%) than UPEC isolates from non-recurrent cases (84.9%) (p<0.004; S3 Fig). Therefore, in this experiment we only used UPEC isolates that were *fimH* positive. We evaluated *fimH*+ UPEC strains from recurrent canine cases (n = 12) of symptomatic UTIs treated with standard antimicrobial protocols with at least once recurrence within 6 months of the first UTI and compared to matched *fimH* + UPEC strains from non-recurrent cases of symptomatic UTIs. The non-recurrent cases all had well documented follow-up to ensure a second UPEC-associated UTI did not occur within 6 months of the initial UTI. UPEC strains from recurrent cases formed more IBCs (12.8 IBCs/ HPF) as compared to UPEC from non-recurrent cases (5.5 IBCs/HPF) (Fig 6A, p<0.001). UPEC strains from recurrent cases also had a higher intracellular bacterial concentration (7.6 $\log_{10}$CFU/well) then UPEC from non-recurrent cases (4.2 $\log_{10}$CFU/well) (Fig 6B, p<0.004). Representative images of urothelial infections show that UPECs from non-recurrent cases (Fig 6C) form less IBCs then UPECs from recurrent cases (Fig 6D). As all the UPEC strains from recurrent and non-recurrent cases were confirmed *fimH*+, further investigation into the mechanisms behind the increased ability of recurrent UPEC strains to form robust IBCs is warranted as *fimH* is not the only gene involved in IBC formation.

## IBCs in canine urothelial cell culture are tolerant to antimicrobial treatment

Both *in vitro* and *in vivo* studies have shown that UPEC IBCs persist intracellularly despite treatment with antimicrobials [6,45]. In the clinical setting, dogs with recurrent UTIs typically receive first-line antimicrobials, which despite demonstrating *in vitro* susceptibility, may result in persistent infections, and the dog will remain culture-positive. Therefore, we wished to investigate the antimicrobial tolerance of IBC-forming UPECs to antimicrobials commonly used to treat canine UTIs. We infected canine primary urothelial cells from three separate donors (n = 3) with three UPEC isolates (n = 3) known to form robust IBCs and allowed IBCs to develop for 24 hours. Subsequently, we treated with infected urothelial cells cultures with enrofloxacin (ENRO, 150µg/mL), cefpodoxime (POD, 100 µg/mL) and doxycycline (DOX, 50µg/mL) at the Cmax (peak concentration) observed in urine based on canine pharmacokinetic studies of each of these antibiotics [46–48]. All the UPEC clinical isolates used for the urothelial infections were susceptible to these antimicrobials as determined by *in vitro* susceptibility testing (data not shown). We found that no antimicrobial treatment tested could eradicate IBCs considerably (>2 logCFU/mL or >50% of the IBCs/HPF) at concentrations at least 10x the minimum inhibitory concentration (MIC), (No Treatment (NT) 9.1 IBCs/HPF versus enrofloxacin (ENRO) 5.2 IBCs/HPF, cefpodoxime (POD) 7.9 IBCs/HPF, and doxycycline (DOX) 6.7 IBCs/HPF) (Fig 7A). Additionally, we did not appreciate a greater than 2 log reduction in bacterial concentration (NT 5.3 $\log_{10}$CFU/well versus ENRO 3.8 $\log_{10}$CFU/well, POD 4.7 $\log_{10}$CFU/well, and DOX 4.1 $\log_{10}$CFU/well, respectively) (Fig 7B). Representative images showing the presence of IBCs (Fig 7C–7F) despite antimicrobial treatment above the MIC. These results show that, similar to traditional biofilms, IBCs grown in canine urothelial cells are tolerant to antimicrobials and therefore are likely to persist intracellularly in the face of antimicrobial treatment despite both ENRO and DOX penetrating well intracellularly [49–51].

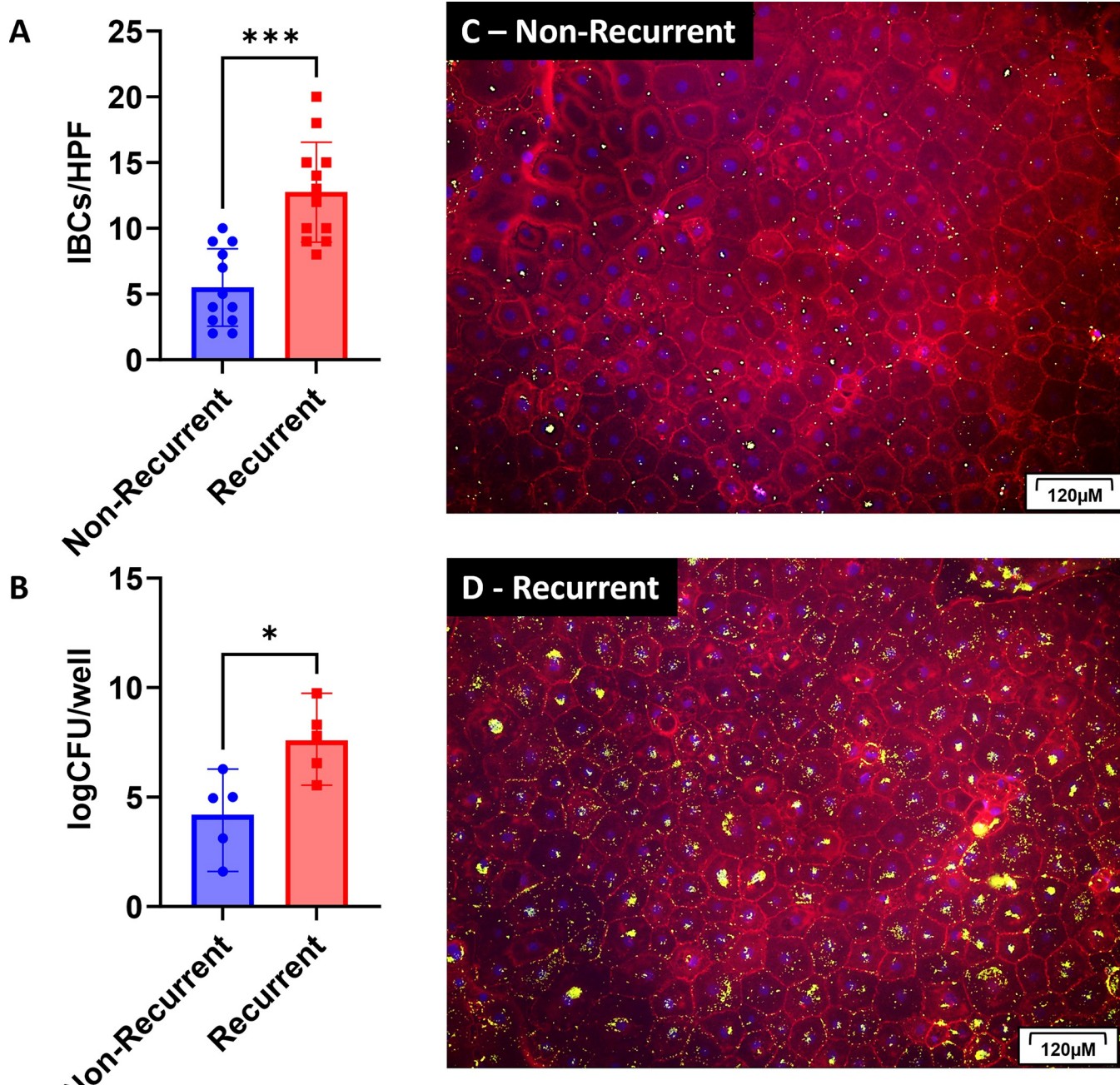

**Fig 6. Canine UPEC isolates from patients with recurrent UTIs form more robust IBCs then isolates from patients with non-recurrent UTIs.** Urothelial cells from canine donors (n = 3) were isolated, cultured, infected, and evaluated as shown in Fig 4. Specifically, canine urothelial cells were infected with canine *fimH*+ UPEC isolates from cases with a well-documented recurrence of the same strain within 6 months of the initial UTI (n = 12). We matched these UPEC isolates with *fimH*+ isolates from non-recurrent cases with well document follow-up in their medical records to ensure that a second UPEC-causing UTI did not occur within 6 months of the initial episode (n = 12). (A) The number of IBCs per high powered field (HPF) were counted for each donor/UPEC combination. The graph represents the average number of IBCs for each UPEC isolate counted in each donor and in each replicate. (B) The bacterial concentration was measured by lysing the cells in a second culture that was unfixed to release the intracellular bacteria. Data is represented as a log CFU per well. (C) Representative images of a urothelial cell infection by a non-recurrent UPEC isolate. (D) Representative images of a urothelial cell infection by recurrent UPEC isolate. Bars are means and standard deviations and significant differences (p<0.05) were determined by an unpaired, nonparametric t-test (for IBCs/HPF) and parametric t-test (for $log_{10}$CFU/mL). Statistically significant differences are denoted with asterisks as follows: *p<0.05, **p<0.01, ***p<0.001, and ****p<0.0001.

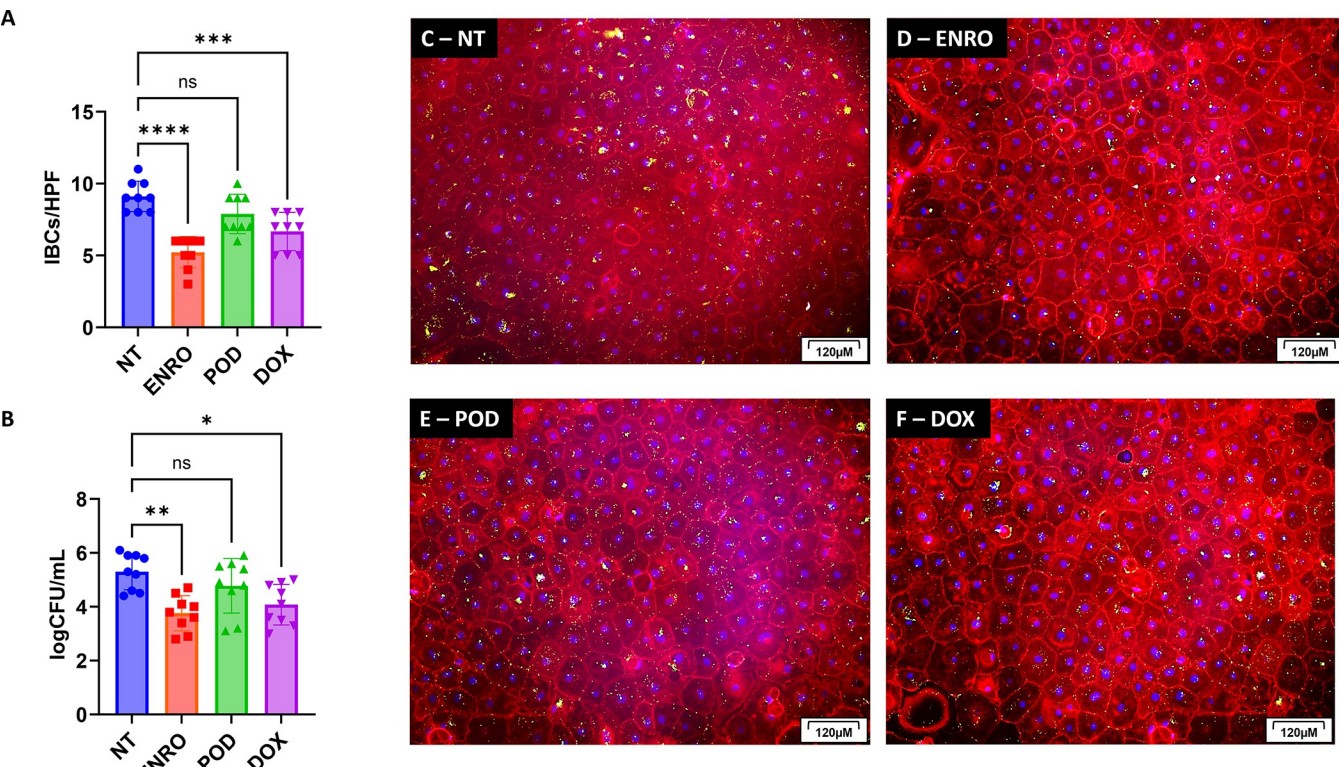

**Fig 7. IBCs formed in canine urothelial cells by canine-sourced UPECs are tolerant to treatment with several antimicrobials from different classes.**
Urothelial cells from canine donors (n = 3) were isolated, cultured, infected and evaluated as in Fig 4. Specifically, canine urothelial cells were infected with canine UPEC isolates known to form robust IBCs based those in Fig 4 (n = 3) and susceptible to the antimicrobials tested herein. (A) The number of IBCs per high powered field (HPF) were counted for each donor/UPEC/treatment combination. The graph represents the average number of IBCs for each UPEC isolate counted in each donor and in each replicate. (B) The bacterial concentration was measured by lysing the cells to release the intracellular bacteria. Data is represented as the average log CFU per well. (C) Representative images of an untreated urothelial cell infection. (D) Representative images of urothelial cell infection treated with 150 μg/mL enrofloxacin (ENRO). (E) Representative images of urothelial cell infection treated with 50 μg/mL cefpodoxime (POD). (F) Representative images of urothelial cell infection treated with 50 μg/mL doxycycline (DOX). Bars are means and standard deviations. Data was analyzed by an ANOVA with Tukey post-hoc. Statistical differences ($p < 0.05$) between groups are indicated by differing letters. Statistically significant differences are denoted with asterisks as follows: *$p < 0.05$, **$p < 0.01$, ***$p < 0.001$, and ****$p < 0.0001$.

## Discussion

[28–30,33–35] Urinary tract infections (UTI) are one of the most common bacterial infections in both humans and dogs [1,52]. UPEC account for >75% of human UTIs [28]. We found that UPEC accounted for 40% of canine urinary tract infections in our retrospective analysis. Although UPEC prevalence was lower, UPEC is still the most frequent uropathogen causing UTIs in dogs. Recurrent infections are a major concern during the treatment and management of UTIs in both dogs and humans [4,53]. [54] We found a slightly higher recurrence rate of 42% in dogs when compared to huams [28], which may be due to the case selection: all cases were from a referral hospital and not a primary care clinic. UPEC account for most human recurrent infections associated with the same UPEC strain in >65% of human recurrent UTIs [32,54]. Similarly, in dogs we found that UPEC accounted for the majority (52%) of recurrent cases. In recent studies, multi-drug resistant UPEC strains are identified in >60% of human isolates [55,56]. We found that 45% of canine UPEC isolates were resistance to at least 3 classes of antimicrobials. In addition, extended spectrum beta-lactamase production (ESBL) has been identified in >30% of human UPEC isolates [35,55]. In our canine population, we observed 33% of canine UPEC isolates harboring one or more beta-lactamase or ESBL-associated genes.

In both human and murine UTI studies, the formation of intracellular bacterial communities (IBCs) within the urothelium is a key attribute of UPEC pathogenesis [57]. UPEC readily attach, invade, and colonize urinary epithelial cells [11,58] and this ability is often implicated in recurrent UTIs [59,60]. A study in mice showed UPEC persisting within the bladder epithelium and causing recurrence in one third of the mice [61]. A publication from Rosen et al. 2007 was the first to show IBCs within sloughed urothelial cells from urine of women with UTI symptoms [27]. In that study, 1 out of 5 urine samples had evidence of IBCs and filamentous bacteria in 50% of the urine samples. We found evidence of IBC formation and filamentous bacteria in 60% and 77% of urine cytologies from UPEC-associated UTIs, respectively. We speculate that the higher rates of IBCs and filamentous bacteria observed in our canine study was because we only evaluated dogs with a positive UPEC culture while Rosen et al., 2007 evaluated UTIs caused by a variety of uropathogens. Other uropathogens such as *Klebsiella pneumoniae* and *Proteus mirabilis* can also form IBCs, but no evidence of IBC formation in gram-positive species has been reported [62–64]. We also found evidence of IBCs within the urothelium of the bladder of dogs with a positive UPEC culture and UTI symptoms. These IBCs were similar in structure and appearance to the IBCs seen within murine urothelium during experimental UPEC infection [11,27].

Others have shown that human primary urothelial cells, human urothelial cells lines and bladder-on-chip models are able to support IBC formation by UPECs [39,45,65,66]. Therefore, we hypothesized that canine urothelial cells might be able to support IBC formation as well. Indeed, we demonstrated that most UPEC isolates were able to form IBCs in our *in vitro* canine urothelial cell model. These results correlate well with a murine study showing that most human derived UPECs could form IBCs in a murine *in vivo* UTI model [12,61,66].

The ability of UPEC to adhere to and invade into the urothelium to form IBCs is facilitated by the type 1 pilus-associated adhesin FimH, a protein encoded by the *fimH* gene, and which binds to mannosylated uroplakin-proteins that line the apical surface of urothelial cells in the bladder, triggering bacterial internalization about 3-hours post-infection [12,16,37,38]. Other studies in mice have shown that UPEC lacking the *fimH* gene are unable to form IBCs [12,42]. Therefore, we investigated the ability of canine derived UPEC harboring or lacking the *fimH* gene to from IBCs in our canine urothelial cell model. We found that UPEC isolates lacking the *fimH* gene formed fewer IBCs than those isolates with harboring the *fimH* gene. However, in those UPEC isolates harboring the *fimH* gene we observed a range of IBCs from rare to robust.

We also found that UPECs isolated from recurrent UTI cases formed more IBCs then those UPEC isolates from non-recurrent cases. Other studies have shown that certain SNPs within the *fimH* gene can results in a gain or loss of function in binding to urothelial cells [43,44,67]. If UPEC are unable to bind to the urothelial cell, they are subsequently unable to invade into the cell to form IBCs. Therefore, we hypothesize that this variation and association with recurrence could be due to the FimH type or SNPs within the *fimH* gene resulting from pathoadaptation to the urinary environment which will be a line of inquiry for future work.

UPEC within IBCs are protected from killing by antimicrobials [6,17]. This protection arises from both decreased penetration of the antimicrobial into the bacterial community within the urothelial cell and the reduced metabolic state of bacteria within the IBC [65,68,69]. We infected canine urothelial cells with UPEC strains know to form robust IBCs from previous screening for 24 hours prior to treatment with a variety of antimicrobials from different drug classes used to treat canine UTIs [70,71]; specifically, the fluoroquinolone enrofloxacin, the cephalosporin cefpodoxime, and the tetracycline doxycycline. We challenged the UPEC-infected urothelial cells with concentrations achievable in canine urine after standard dosing of each antimicrobial [72–74]. None of the antimicrobials at clinically relevant concentrations

were able to reduce the number of IBCs or the bacterial load greater than 2-log; although some variability was observed in response, this is likely due to intracellularly achievable concentration differences between antimicrobial classes. Our outcomes are supported by several studies showing that IBCs are tolerant to antimicrobial treatment [6,17,75]. For example, in a murine bladder infection model treatment with trimethoprim-sulfamethoxazole did not eradicate bacteria within the bladder [17]. Another study showed persistence of UPEC within the bladder epithelium using multiple antibiotics with urine concentration well above the therapeutic threshold of planktonic (single cell) UPEC [6]. In a bladder on chip model, treatment with a beta-lactam, ampicillin, did not reduce the IBC size or distribution within the tissue [45]. These findings correlate with the clinical observations that approximately 14% of women require a second antibiotic prescription within 28 days of diagnosis [75].

In conclusion, we found that the dog is a robust translational model for human UTIs. A recent review article calls for the development of new animal models that better recapitulate human disease [76]. As the dog is a naturally occurring model of UPEC UTIs it holds promise as a more clinically relevant model then the induced murine or porcine models currently in use [77–81]. The results of this study show that the pathogenesis of canine UPEC UTIs, with the formation of antimicrobial tolerant *fimH* dependent IBCs, are similar to human UTIs caused by UPECs. Importantly, the use of the dog would serve as a One Health model as results from these studies could directly apply to both human and veterinary medicine. However, future work is needed that expands upon the presented in this manuscript into an *in vivo* experimental model that can be used to study pathogenesis and develop novel therapeutics. In addition, direct genomic comparisons of canine and human UPECs could be useful in further assessing the canine UTI model.

## Methods

### Bacterial strains

Clinical UPEC isolates (n = 206) were obtained from routine clinical aerobic cultures of canine urine submitted to the North Carolina State University's College of Veterinary Medicine (NCSU-CVM), Microbiology and Molecular Diagnostics Laboratory (Raleigh, NC). Isolates were included if 1) *E. coli* was the sole etiological agent (monospecies) 2) urine samples were collected via cystocentesis 3) the quantity of *E. coli* recovered was greater than 100,000 CFU/ mL as determined by plating of a standard urine volume and 4) the dog was considered to have a UTI based on clinical presentation and diagnosis by the attending veterinarian. All isolates were confirmed as *E. coli* using matrix assisted laser desorption/ionisation time of flight mass spectrometry (MALDI-TOF) (Vitek MS, Biomerieux; Marcy-l'Étoile, France). Isolates were saved as frozen stocks on glycerol at -80°C. Antimicrobial susceptibility testing was performed on all isolates using the broth microdilution method by a commercial system for the clinical report (Sensititre™; ThermoFisher Scientific, Waltham, MA) using the Companion Animal MIC Plate (Companion Animal MIC Plate). The minimum inhibitory concentrations (MIC) of each antimicrobial reagent were classified as susceptible (including intermediate) or resistant based on the Clinical & Laboratory Standards Institute (CLSI) breakpoints when available [82]. Isolates were characterized for the following UPEC-associated virulence genes attributed to extraintestinal *E. coli* (ExPEC): *cnf-1*, *hlyA*, *fimH* and *papG* using previously described PCR assays [26,83].

### Recurrent UTI classification

Medical records of the North Carolina State University (Raleigh, NC) Veterinary Teaching Hospital (VTH) were searched to identify UPEC isolates from dogs that were diagnosed with

recurrent UTIs. Isolates were designated as recurrent isolates based on at least ≥2 UTIs confirmed by positive aerobic bacterial cultures of *E. coli* during a 6-month period.

## Clinical specimens

Urine cytologies and bladder biopsies were sourced from NCSU-CVM Clinical Pathology & Immunology or Histopathology Laboratories (Raleigh, NC). Cytologies and biopsies were chosen for evaluation if the dog had a concurrent positive *E. coli* monoculture by NCSU-CVM's Clinical Microbiology and Molecular Diagnostics Laboratory, presented with a symptomatic UTI, and had evidence of bacteria by Diff-Quick staining for cultures as assessed by a qualified clinical pathology technician and evidence of inflammation and infection on histology as assessed by a board-certified clinical pathologist. All cytology and biopsy specimens would have been submitted to the laboratories as part of the clinical and diagnostic work-up at the discretion of the treating veterinarian and were not submitted for inclusion in this study. Owners are notified that the remaining sample can be used for research purposes during the patient intake.

Cytologies and biopsies were then assessed for the presence of IBCs by immunofluorescence. For cytologies (n = 30), prepared cytospin slides were fixed with 2% paraformaldehyde, washed with sterile phosphate-buffered saline (PBS) and permeabilized with 1% Bovine Serum Albumin (BSA)/0.3% Triton X-100 for 1 hour at room temperature and blocked for 1 hour with 5% BSA. Slides were then incubated overnight with Rabbit Anti-*E. coli* FITC antibody (1:100, ab30522, Abcam, Cambridge, UK) and Mouse Anti-Uroplakin III primary antibody (1:300, ab78196, Abcam, Cambridge, UK). Slides were washed three times in sterile PBS and incubated with the secondary antibody Goat Anti-Mouse IgG H&L Alexa Fluor® 647 (1:1000, ab150115, Abcam, Cambridge, UK) for 1 hour. Slides were then washed again with sterile PBS and mounted with DAPI Fluoromount-G (Southern Biotech, Birmingham, AL). For biopsies (n = 10), paraffin-embedded tissue was deparaffinized with Histo-Clear II® (Electron Microscopy Sciences, VWR International, Radnor, PA) followed by ethanol dehydration and heat induced antigen retrieval. Slides were then permeabilized in 1% BSA/0.3% Triton X-100 for 10 minutes and blocked for 1 hour with 5% BSA. Slides were then incubated overnight with Rabbit Anti-*E. coli* FITC antibody (1:100, ab30522, Abcam, Cambridge, UK) and Rabbit Anti-E Cadherin (1:250, ab15148, Abcam, Cambridge, UK). Slides were washed three times in sterile PBS and the secondary antibody Goat Anti-Rabbit IgG H&L Alexa Fluor® 555 (ab150078) (1:1000, ab150115, Abcam, Cambridge, UK) for 1 hour. Slides were then washed again with sterile PBS and mounted with DAPI Fluoromount-G (Southern Biotech, Birmingham, AL). Stained cytologies and biopsies were imaged on an Olympus IX73 inverted scope with DP80 camera using appropriate fluorescent channels (Olympus Corporation, Shinjuku, Tokyo, Japan).

## Canine urothelial cell isolation

Canine bladders (n = 10) were aseptically collected postmortem from apparently healthy dogs after planned euthanasia by a North Carolina animal control facility. The dogs were euthanized by the local animal control facility based on unfit temperament for adoption at the discretion of the facility; the authors had no role in the decision to euthanize and were notified of the planned euthanasia prior to attending. No dogs were euthanized for the purpose of this study. Dogs were apparently healthy, in good body condition and did not show any signs of urinary or renal disease. At the time of collection, the distal abdomen was aseptically prepped, and a surgical incision was made to reach the urinary bladder. A urine sample was collected for culture after surgical incision by cystocentesis prior to removing the bladder. The bladder was removed by severing of the ureters and ligaments and excising the bladder at the neck

before placing the bladder in PBS containing antibiotic-antimycotic (Gibco® Antibiotic-Anti-mycotic ThermoFisher Scientific, Waltham, MA). Bladders were transported to the laboratory on ice and all subsequent work was completed in a biosafety cabinet. The bladder was turned inside out to expose the mucosa. Mucosa was evaluated for any noticeable gross pathology. If a bladder had a positive urine culture or the mucosa appeared grossly abnormal, the bladder or any isolated cells were excluded from further use. The mucosa neck of the bladder was sutured closed to ensure only mucosa was exposed. The tissue was then suspended in an Erlenmeyer flask in HBSS containing antibiotic-antimycotic and 2.4U/mL Gibco® Dispase II (Thermo-Fisher Scientific, Waltham, MA) and incubated for 2 hours at 5% CO2, 90% humidity, and 37˚C under constant rotation (S1A Fig). After 2 hours, any remaining tissue on the mucosal surface was gently scraped off and added to the dispersed cells, passed through a 100μm filter, and centrifuged at 300g for 10 min. The cell pellet was washed and resuspended in fresh HBSS and a live cell count was determined using a Cellometer® Auto 2000 and ViaStain™ AOPI Staining Solution (Nexcelom Bioscience LLC, Lawrence, MA). Cells were then seeded onto Nunc™ Collagen I Coated EasYFlasks™ (ThermoFisher Scientific, Waltham, MA) at 40,000 cells/cm$^2$ in Urothelial Cell Proliferation Media (UCPM, Gibco™ Defined Keratinocyte SFM containing Defined Keratinocyte-SFM Growth Supplement and antibiotic-antimycotic) and incubated overnight at 5% CO2, 90% humidity, and 37˚C. Dead or non-adherent cells were removed the next day by washing with warm PBS three times and adding fresh UCPM. The culture medium was changed every 48 hours for 5–7 days. UCPM is a serum-free media and was used to proliferate urothelial cells while decreasing fibroblast contamination; by the end of 5–7 days, depending on the donor, a pure urothelial cell culture was appreciated by brightfield microscopy and analysis of cellular morphology as described by others (S1B Fig) [40,84,85]. When cultures reached a confluency of 75–95% pure morphologically epithelial-like cells, cells were detached from the growth surface with the use of Invitrogen™ Accutase (0.12 mL/cm$^2$) (ThermoFisher Scientific, Waltham, MA). Cells were counted again as above and frozen in aliquots of 1×10$^6$ cells/mL in liquid nitrogen until future use.

## Canine urothelial cell infection and antimicrobial treatment

Canine urothelial cells were thawed and seeded at 40,000 cells/cm$^2$ onto collagen coated 24-well tissue culture plates (Nunc™ Collagen I Coated Multidishes, ThermoFisher Scientific, Waltham, MA) in UCPM for 24 hours. Dead or non-adherent cells were removed the next day by washing with warm PBS three times and adding fresh UCPM. Cells were grown for another 48 hours, or until 75–95% confluence was reached. Thereafter, the culture media was changed to promote urothelial cell differentiation for expression of uroplakins required for UPEC pathogenesis as described elsewhere[41,86,87]. Briefly, media was changed to Urothelial Cell Differentiation Media (UCDM, 3 parts Dulbecco's modified Eagle's medium to 1-part F12 containing 5% fetal bovine serum, antibiotic-antimycotic and 2mM calcium). Cells were cultured in UCDM for 48 hours, followed by a media change, and then cultured for an additional 48 hours. We have shown that this time frame allows for optimal uroplakin expression in our model (S1C Fig). Thereafter, urothelial cells (n = 3 biological replicates per experiment) were infected with various individual UPEC strains (n = 30) described above in "Bacterial strains" at a multiplicity of infection (MOI) of 10:1 for 3 hours at 37˚C and 5% CO$_2$. After 3 hours, cells were washed five times with sterile PBS and incubated with media containing 100 μg/ml gentamicin to kill any extracellular bacteria. Infections were carried out for 21 hours after the gentamicin treatment. For antimicrobial challenge experiments, 24 hours post-infection, the gentamicin containing-media was removed, and the cellular monolayer was washed five times with sterile PBS before media containing the appropriate antimicrobial was added for an

additional 24 hours. Enrofloxacin (ENRO), cefpodoxime (POD) or doxycycline (DOX) were used at 150, 50 and 50 μg/mL respectively which reflect achievable canine urine concentrations [46–48].

## Intracellular bacterial community (IBC) analysis

IBCs were assessed by two methods. First, the intracellular bacterial load was assessed in each culture by lysing urothelial cells and performing serial dilutions and plate counting to determine intracellular CFU [39]. Second, we visualized and quantified IBCs by immunofluorescence. Briefly, after infection and/or treatment, cultures were washed five times with PBS and fixed with 2% paraformaldehyde. Cultures were then permeabilized with 0.3% Triton X-100 and blocked with 5% BSA prior to staining with Rabbit Anti-*E. coli* FITC (1:100, ab30522, Abcam, Cambridge, UK) and Alexa Fluor™ 555 Phalloidin according to manufacturer's directions (ThermoFisher Scientific, Waltham, MA). Nuclei were stained with DAPI Fluoromount-G® (SouthernBiotech, Birmingham, AL) prior to imaging on an Olympus IX73 inverted scope with DP80 camera using appropriate fluorescent channels (Olympus Corporation, Shinjuku, Tokyo, Japan). To quantify the number of IBCs by visualization, the average number of IBCs observed per high powered field (HPF) were counted; five HPFs at 10x magnification were counted per well. A secondary quantification method was performed to determine the intracellular bacterial load. Using this method, urothelial cell monolayers were washed 3x with PBS and lysed with 0.1% Triton X for 10 minutes. After lysis, serial dilutions were performed, and spot plated in triplicate at each serial dilution for colony counts.

## Statistical analysis

Data in graphs is presented as bars representing the mean and standard deviations. Data was analyzed for normality by the Shapiro-Wilk test. Based on normality, single data points such as IBCs/HPF and $\log_{10}$ CFU/mL were analyzed by either a 1-way non-parametric (Kruskal-Wallis Test) or unpaired t-test. A 2-way ANOVA with Tukey's post hoc test was used to analyze the antimicrobial effects on IBCs. Frequencies were evaluated by a chi-square test. Sex and other covariates were included as fixed effects in the model if indicated. Analysis was performed using JMP Pro 11.0 software (SAS Institute Inc., Cary, NC). All graphs were generated using GraphPad Prism (GraphPad Software Inc., La Jolla California USA). For all comparisons, p<0.05 was considered statistically significant.

## Supporting information

**S1 Fig. Canine urothelial cell isolation and differentiation.** Canine bladders (n = 10) were aseptically collected postmortem from apparently healthy dogs after planned euthanasia by an animal control facility. (A) The bladder was removed, turned inside out to expose the mucosa and suspended in an Erlenmeyer flask under constant rotation. The tissue was treated with Dispase to remove the urothelial cells and cells were subsequently seeded onto collagen coated flasks and grown in culture. (B) By 7 days a pure urothelial cell culture was appreciated by brightfield microscopy (10x) and analysis of cellular morphology as described by others [40,84,85,88]. Urothelial cells were frozen in aliquots and stored in liquid nitrogen until future use. (C) Canine urothelial cells were thawed, and seeded collagen coated tissue culture plates and grown until confluency. Thereafter, the culture media was changed to promote urothelial cell differentiation for expression of uroplakins required for UPEC pathogenesis as described elsewhere[41,86,89]. Expression of various uroplakins was determined by real-time polymerase chain reaction using canine specific primers and the housekeeping gene GAPDH. Relative

expression compared to the day 0 (before changing to UPDM) is presented at $2^{-\Delta\Delta Ct}$.
(TIF)

**S2 Fig. Size of IBCs within Canine Urothelial Cells.** Intracellular bacterial communities were evaluated via immunofluorescence microscopy. (A) Canine urothelial cells were infected with canine specific UPECs designated by a strain number. Ten IBCs per strain were measured. The minimum size of the communities was observed as 10uM with the maximum size being 85uM with an average of 29μm. The average of the length and width was taken to denote size as these images were taken in two dimensions not three. (B) Examples of cells with IBCs (white arrows) that were counted and examples of cells with intracellular bacteria but not IBCs that were not counted (blue arrows).
(TIF)

**S3 Fig. Canine UPEC isolates from recurrent UTIs correlate with *fimH* gene presence but not multidrug resistance (MDR).** Canine UPEC isolates (n = -206) were classified as recurrent with well-documented recurrence of the same phenotypic strain within 6 months of the initial UTI. (A) Each isolate was evaluated for the presence or absence of the *fimH* gene by PCR. (B) Incidence of multidrug resistance was defined by resistance to greater than or equal to three antimicrobial drug classes. Bars are means and standard deviations and significant differences (p<0.05) were determined by an unpaired, nonparametric t-test. Statistically significant differences are denoted with asterisks as follows: *p<0.05, **p<0.01, ***p<0.001, and ****p<0.0001.
(TIF)

## Acknowledgments

The authors would like to thank Ms. Jasmin Huang and Ms. Melissa Byrd from the Clinical Microbiology Laboratory at North Carolina State University's College of Veterinary Medicine (NCSU-CVM), Ms. Laura Miller from the Histology Laboratory at NCSU-CVM, Ms. Lynnette McCall from the Clinical Pathology Laboratory at NCSU-CVM, and Ms. Cathryn Hubbard from the Clinical Skills Laboratory at NCSU-CVM.

## Author Contributions

**Conceptualization:** Shelly L. Vaden.

**Data curation:** Jessica M. Gilbertie.

**Formal analysis:** Jessica M. Gilbertie, Breanna J. Sheahan, Megan E. Jacob.

**Investigation:** Jessica M. Gilbertie, Megan E. Jacob.

**Methodology:** Jessica M. Gilbertie, Breanna J. Sheahan, Megan E. Jacob.

**Supervision:** Shelly L. Vaden, Megan E. Jacob.

**Visualization:** Breanna J. Sheahan.

**Writing – original draft:** Jessica M. Gilbertie.

**Writing – review & editing:** Jessica M. Gilbertie, Breanna J. Sheahan, Shelly L. Vaden, Megan E. Jacob.

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
