## [Decision Letter · Decision Letter 0]

28 Dec 2022

PONE-D-22-30262Canine urothelial cell model to study intracellular bacterial community development by uropathogenic Escherichia coliPLOS ONE

Dear Dr. Jacob,

Thank you for submitting your manuscript to PLOS ONE. After careful consideration, we feel that it has merit but does not fully meet PLOS ONE’s publication criteria as it currently stands. Therefore, we invite you to submit a revised version of the manuscript that addresses the points raised during the review process. Please submit your revised manuscript by Feb 11 2023 11:59PM. If you will need more time than this to complete your revisions, please reply to this message or contact the journal office at plosone@plos.org. Please include the following items when submitting your revised manuscript:A rebuttal letter that responds to each point raised by the academic editor and reviewer(s). You should upload this letter as a separate file labeled 'Response to Reviewers'.A marked-up copy of your manuscript that highlights changes made to the original version. You should upload this as a separate file labeled 'Revised Manuscript with Track Changes'.An unmarked version of your revised paper without tracked changes. You should upload this as a separate file labeled 'Manuscript'.

We look forward to receiving your revised manuscript.

Kind regards,

Kwame Kumi Asare, Ph.D

Academic Editor

PLOS ONE

Journal Requirements:

2. We noticed that animals from a control facility were used as study subjects in this research. According to PLOS ONE animal ethics policies, manuscripts describing studies that use random source, shelter, or stray animals are subject to additional scrutiny and may be rejected if sufficient ethical and scientific justification for the study design is lacking (https://journals.plos.org/plosone/s/submission-guidelines#loc-animal-research).

In your methods section, please provide scientific and ethical justification for the use of these animals in the specific study design, and please explain whether other animals could have been used or not instead.

Thank you very much for your attention to our requests.

3. Please ensure that you have specified (1) whether consent was informed and (2) what type you obtained (for instance, written or verbal, and if verbal, how it was documented and witnessed). If your study included minors, state whether you obtained consent from parents or guardians. If the need for consent was waived by the ethics committee, please include this information.

4. PLOS requires an ORCID iD for the corresponding author in Editorial Manager on papers submitted after December 6th, 2016. Please ensure that you have an ORCID iD and that it is validated in Editorial Manager. To do this, go to ‘Update my Information’ (in the upper left-hand corner of the main menu), and click on the Fetch/Validate link next to the ORCID field. This will take you to the ORCID site and allow you to create a new iD or authenticate a pre-existing iD in Editorial Manager. Please see the following video for instructions on linking an ORCID iD to your Editorial Manager account: https://www.youtube.com/watch?v=_xcclfuvtxQ.

Reviewers' comments:

Reviewer's Responses to Questions

**Comments to the Author**

1. Is the manuscript technically sound, and do the data support the conclusions?

Reviewer #1: Yes

Reviewer #2: Yes

2. Has the statistical analysis been performed appropriately and rigorously? 

Reviewer #1: No

Reviewer #2: Yes

3. Have the authors made all data underlying the findings in their manuscript fully available?

Reviewer #1: Yes

Reviewer #2: Yes

4. Is the manuscript presented in an intelligible fashion and written in standard English?

Reviewer #1: Yes

Reviewer #2: Yes

5. Review Comments to the Author

Reviewer #1: In this paper, Gilbertie and colleagues describe a series of experiments identifying intracellular uropathogenic E. coli (UPEC) in the urothelial tissues of dogs with UTIs, as well as the capability of canine uroepithelial cells to be propagated and to support an intracellular, IBC-like state for UPEC. Several suggestions for improvement are given below.

Major points:

1. The Introduction could do a better job of specifying what is and what is not known about canine UTI prior to the present study. In the passage starting on line 69, the authors state that “dogs have similar UTI pathogenesis” to humans and that dogs have similar rates of drug-resistant uropathogens, giving multiple references for these statements. Then, in the subsequent paragraph, the authors state that the goal of the present study was to “determine if dogs demonstrate similar UTI pathogenesis” to humans, and that the present paper “demonstrated that dogs are clinically affected by similar uropathogens with similar resistance rates…” On a related note, the authors state (line 70) that dogs have “urinary biology comparable to humans” but this statement needs more specifics. What aspects of urinary biology are similar between the two species?

2. The letter-labels for the panels in Figure 1 don’t match between the legend (lines 116-128) and the figure itself.

3. In the paragraph starting on line 250, it appears that 12 recurrent UTI strains and 12 non-recurrent strains were evaluated. But then “97.5%” and “84.9%” of the strains carry fimH, which presumably reflects n=206 (Figure S2 legend)? The text in this paragraph should be clarified as to how many strains were evaluated for each analysis.

4. In Figure 7, it isn’t clear what is meant by “significant differences are indicated by differing letters” – in other words, what the a’s and b’s mean in panels A and B. Accompanying this, it would be helpful if in the Discussion, the authors commented on these differences and how they might correlate with the likely intracellular penetration of the chosen antibiotics. Specifically, fluoroquinolones (e.g., enrofloxacin) and doxycycline would be expected to penetrate cells better than a beta-lactam (here cefpodoxime), and there appears to be a correlation between this property and the antibacterial effect as measured by CFU and IBC numbers.

5. In Figure S2, the measurements of both fimH carriage (panel A) and MDR (panel B) are binary in nature. So, what do the bars represent here? There should not be standard deviations in this type of data. Accompanying that, one would expect that a chi-square analysis rather than “an unpaired, nonparametric t-test” (line 279) would be appropriate for comparisons.

6. The vertebrate animals information would benefit from additional clarification. The authors state that all the samples were collected for clinical purposes and prior to the start of any analysis (i.e., they were banked). What was the clinical indication for bladder biopsies in dogs with active UTIs?

Minor points:

1. Given that no methodologic details are not fully available in the abstract, one should remove the specific numerical data from the abstract. For example, “4.2 log10 CFU/well” doesn’t have specific meaning as the reader hasn’t yet read the methods.

2. Line 44: “20% of dogs with UTIs will develop a recurrent infection…” during what period of follow-up (in reference 4)?

3. Line 182: Beyond “similar morphology,” can the authors be more specific as to the size of intracellular communities or the apparent number of bacteria, in canine uroepithelial cells vs the immortalized human bladder cell line?

4. Some of the figure legends could be shortened by reducing methodologic detail (these details already appear in the Methods section).

5. The Discussion could probably also be shortened by reducing the degree to which the current Results are recapped.

6. Panels A and B of Figure 2 could probably be eliminated, as they reflect simple percentages that are described in the text.

Reviewer #2: The authors have written an informative manuscript. However, the authors should proofread the manuscript and correct some minor spelling mistakes and strictly follow the journal's author guidelines.

The sentence in line 22 and 23 is incomplete.

The sentence in line 63 and 64 is not well understood.

Please check the spelling of ' their) in line 66.

Please elaborate more on the statistical analysis. Most of the statement have been made in the 'results' section. For example, non-parametric t-test.

The authors did not include control strains in the experimental section

Authors should please take a second look at the clarity of of the labels

Authors should make comments on the ethical approval

6. PLOS authors have the option to publish the peer review history of their article (what does this mean?). If published, this will include your full peer review and any attached files.

Reviewer #1: No

Reviewer #2: **Yes: **Samuel Addo Akwetey

---

## [Author Response · Author response to Decision Letter 0]

12 Sep 2023

We thank the reviewers for their time and thoughtful suggestions on improving this manuscript. There are detailed responses to each concern noted. 

Reviewer #1: In this paper, Gilbertie and colleagues describe a series of experiments identifying intracellular uropathogenic E. coli (UPEC) in the urothelial tissues of dogs with UTIs, as well as the capability of canine uroepithelial cells to be propagated and to support an intracellular, IBC-like state for UPEC. Several suggestions for improvement are given below.

Major points:

1. The Introduction could do a better job of specifying what is and what is not known about canine UTI prior to the present study. In the passage starting on line 69, the authors state that “dogs have similar UTI pathogenesis” to humans and that dogs have similar rates of drug-resistant uropathogens, giving multiple references for these statements. Then, in the subsequent paragraph, the authors state that the goal of the present study was to “determine if dogs demonstrate similar UTI pathogenesis” to humans, and that the present paper “demonstrated that dogs are clinically affected by similar uropathogens with similar resistance rates…” On a related note, the authors state (line 70) that dogs have “urinary biology comparable to humans” but this statement needs more specifics. What aspects of urinary biology are similar between the two species?

We agree that the introduction was confusing, and we appreciate the reviewers point. We have edited the introduction to state that we know dogs have similar UTI epidemiology in that they are infected with similar etiologic agents, are infected by similar UPEC sequence types, have similar MDR and ESBL carriage within their UPEC population and are afflicted by similar recurrence rates. However, if canine and human UTIs have a similar pathogenesis is unknown. Therefore, the goal of this manuscript was to determine if IBCs form in canine UTIs in a similar manner as observed in humans.

2. The letter-labels for the panels in Figure 1 don’t match between the legend (lines 116-128) and the figure itself.

Thank you for noting this mistake. This has now been corrected.

3. In the paragraph starting on line 250, it appears that 12 recurrent UTI strains and 12 non-recurrent strains were evaluated. But then “97.5%” and “84.9%” of the strains carry fimH, which presumably reflects n=206 (Figure S2 legend)? The text in this paragraph should be clarified as to how many strains were evaluated for each analysis.

We edited this experimental description for clarity. 

4. In Figure 7, it isn’t clear what is meant by “significant differences are indicated by differing letters” – in other words, what the a’s and b’s mean in panels A and B. 

Letter designations are given when using a Tukey post hoc for multiple comparisons. Therefore, a bar with an “a” above it is statistically different from a bar with any other letter designations such as “b” or “c” but it not different from another “a” or a bar with an “a,b” above it. We tried to clarify that in the figure legend more.

Accompanying this, it would be helpful if in the Discussion, the authors commented on these differences and how they might correlate with the likely intracellular penetration of the chosen antibiotics. Specifically, fluoroquinolones (e.g., enrofloxacin) and doxycycline would be expected to penetrate cells better than a beta-lactam (here cefpodoxime), and there appears to be a correlation between this property and the antibacterial effect as measured by CFU and IBC numbers.

Good point! We have added this observation to the discussion section; in the interest of discussion length as previously noted by reviewers, we did not go into detail as it was not the primary outcome of interest; we can certainly develop this further if the reviewer feels like it would be important. 

5. In Figure S2, the measurements of both fimH carriage (panel A) and MDR (panel B) are binary in nature. So, what do the bars represent here? There should not be standard deviations in this type of data. Accompanying that, one would expect that a chi-square analysis rather than “an unpaired, nonparametric t-test” (line 279) would be appropriate for comparisons.

We removed the error bars and performed a chi-square test instead of a t-test. The figure has been updated to represent the edits.

6. The vertebrate animals information would benefit from additional clarification. The authors state that all the samples were collected for clinical purposes and prior to the start of any analysis (i.e., they were banked). What was the clinical indication for bladder biopsies in dogs with active UTIs?

Bladder biopsies are taken during uroscopy at the discretion of the treating veterinarian. The biopsies are sent to the pathology service to evaluate the bladder tissue for the presence of inflammation, infection, or neoplasm. These samples are banked and can be used for research purposes. We have tried to clarify this in the manuscript by adding that these are diagnostic specimens submitted for the management of a patient and not because of inclusion in the study. Still, all samples submitted to the NCSU CVM are approved to be used for research purposes. 

Minor points:

1. Given that no methodologic details are not fully available in the abstract, one should remove the specific numerical data from the abstract. For example, “4.2 log10 CFU/well” doesn’t have specific meaning as the reader hasn’t yet read the methods.

This numerical data has been removed from the abstract per this reviewer’s request.

2. Line 44: “20% of dogs with UTIs will develop a recurrent infection…” during what period of follow-up (in reference 4)?

This has been updated.

3. Line 182: Beyond “similar morphology,” can the authors be more specific as to the size of intracellular communities or the apparent number of bacteria, in canine uroepithelial cells vs the immortalized human bladder cell line?

Not at this stage but this would be a good idea for future work.

4. Some of the figure legends could be shortened by reducing methodologic detail (these details already appear in the Methods section).

We have followed this suggestion and reduced the figure legends.

5. The Discussion could probably also be shortened by reducing the degree to which the current Results are recapped.

We have followed this advice and removed paragraph 1 that recapped the results.

6. Panels A and B of Figure 2 could probably be eliminated, as they reflect simple percentages that are described in the text.

We prefer to leave panels A and B in Figure 2 in but will leave this to the discretion of the editor.

Reviewer #2: The authors have written an informative manuscript. However, the authors should proofread the manuscript and correct some minor spelling mistakes and strictly follow the journal's author guidelines.

The sentence in line 22 and 23 is incomplete. Corrected.

The sentence in line 63 and 64 is not well understood. We have edited this sentence.

Please check the spelling of ' their) in line 66. Corrected.

Please elaborate more on the statistical analysis. Most of the statement have been made in the 'results' section. For example, non-parametric t-test. We have added to the statistical analysis section with statements from our figures/results sections.

The authors did not include control strains in the experimental section. We did not use any control strains only clinical UPEC strains from canine UTIs.

Authors should please take a second look at the clarity of the labels. We have double checked the labels and corrected where appropriate. Most of this clarification was addressed through other reviewer comments. 

Authors should make comments on the ethical approval. We have added further comments on the ethical approval.

---

## [Decision Letter · Decision Letter 1]

31 Oct 2023

PONE-D-22-30262R1Canine urothelial cell model to study intracellular bacterial community development by uropathogenic Escherichia coliPLOS ONE

Dear Dr.Megan,

Thank you for submitting your manuscript to PLOS ONE. After careful consideration, we feel that it has merit but does not fully meet PLOS ONE’s publication criteria as it currently stands. Therefore, we invite you to submit a revised version of the manuscript that addresses the points raised during the review process.

We look forward to receiving your revised manuscript.

Kind regards,

Kwame Kumi Asare, Ph.D

Academic Editor

PLOS ONE

Reviewers' comments:

Reviewer's Responses to Questions

**Comments to the Author**

1. If the authors have adequately addressed your comments raised in a previous round of review and you feel that this manuscript is now acceptable for publication, you may indicate that here to bypass the “Comments to the Author” section, enter your conflict of interest statement in the “Confidential to Editor” section, and submit your "Accept" recommendation.

Reviewer #1: All comments have been addressed

Reviewer #3: (No Response)

Reviewer #4: (No Response)

Reviewer #5: All comments have been addressed

Reviewer #6: (No Response)

Reviewer #7: All comments have been addressed

Reviewer #8: (No Response)

Reviewer #9: (No Response)

Reviewer #10: All comments have been addressed

Reviewer #11: (No Response)

2. Is the manuscript technically sound, and do the data support the conclusions?

Reviewer #1: Yes

Reviewer #3: No

Reviewer #4: Yes

Reviewer #5: Yes

Reviewer #6: No

Reviewer #7: Yes

Reviewer #8: Yes

Reviewer #9: Partly

Reviewer #10: Yes

Reviewer #11: No

3. Has the statistical analysis been performed appropriately and rigorously? 

Reviewer #1: Yes

Reviewer #3: Yes

Reviewer #4: Yes

Reviewer #5: Yes

Reviewer #6: Yes

Reviewer #7: Yes

Reviewer #8: Yes

Reviewer #9: Yes

Reviewer #10: Yes

Reviewer #11: No

4. Have the authors made all data underlying the findings in their manuscript fully available?

Reviewer #1: Yes

Reviewer #3: Yes

Reviewer #4: Yes

Reviewer #5: Yes

Reviewer #6: No

Reviewer #7: Yes

Reviewer #8: No

Reviewer #9: Yes

Reviewer #10: Yes

Reviewer #11: Yes

5. Is the manuscript presented in an intelligible fashion and written in standard English?

Reviewer #1: Yes

Reviewer #3: Yes

Reviewer #4: Yes

Reviewer #5: Yes

Reviewer #6: No

Reviewer #7: Yes

Reviewer #8: Yes

Reviewer #9: Yes

Reviewer #10: Yes

Reviewer #11: Yes

6. Review Comments to the Author

Reviewer #1: All comments addressed

This field requires 100 character minimum, so I am adding this sentence to reach that goal

Reviewer #3: Dear Author

Thank you for your manuscript submission. I believe that your study has fundamental problem. As you know, E.coli has a powerful arsenal including a wide range of virulence factors. You can not draw a conclusion on the basis of a virulence factor like FimH or other virulence factor in solo regarding a feature like IBC. There are many paradox in your manuscript which each part deny the other. Hence, my decision regarding the present manuscript is Definite Rejection.

Reviewer #4: The only concern that has no been addressed is related with the microscope technique selected for visualization and quantification of IBC. Epifluorescence do not allow to differentiate if a bacteria is on the surface of the cell or inside because of the type of images it gave. In this cases, confocal microscopy is the best option as it is possible to take optical images in the z-plane. Confocal microscopy will confirm that the bacteria are really inside the cell and not in the surface.

Reviewer #5: Article is written in standard English.

Article is acceptable in this form.

All reviewers comments had been addressed.

Reviewer #6: Please ensure the manuscript is submitted as per guidelines of the journal. This is interesting study, and hard work has been done to provide ne information, but there is significantly change needed to write in research manuscript format. Specific comments are provided in the manuscript. The author has made change as requested from previous review. However, the manuscript is not up to the standard of journal. The method section has all previous work description, and some current work description. Result section has method and discussion.

The ANOVA analysis performed, and did not mention the result. For what section did you performed ANOVA? Please elaborate. The figure still needs clear caption. Results for each experiment conducted in this study would added value in terms of data sharing as there are only 9 sample size. Figure with IBCs and bar graph do not provide enough data to researcher who want to replicate the study in future.

Reviewer #7: Authors have performed a detailed study and manuscript is well written.

Authors have written in the Ethics statement that IACUC approval was not required. I think this statement should be revisited.

Reviewer #8: This study is aimed to investigate whether dogs evidence similar UPEC-associated UTI pathogenesis with IBC formation, and to validate an in vitro IBC model using primary canine urothelial cells.

The work overall is solid, although I do have some comments (noted below).

Comments

- In all figures, font types are unreadable, please modify. Similar problems arise with the scale bar in most of the micrographs.

- Information regarding the retrospective study performed should be added in Materials and Methods Section, including its ethical approval.

- L334. Data regarding antimicrobial susceptibility should be shown.

- Authors should proofread the manuscript and correct some minor spelling mistakes.

Reviewer #9: The authors described the capability of uropathogenic Escherichia coli strains, isolated from canine UTIs, to invade bladder cells and to form the intracellular bacterial communities (IBCs). This represents a very interesting topic for microbiologists characterizing UPEC strains and the paper provides a basis to study the genotypic and phenotypic relationship between human and dog associated UPEC. Images of IBCs are solid and results reported in the manuscript are quite in line with the published data. However, authors should address the following points:

Minor points:

Introduction section should be improved

Lines 44-47 Is there any genetic/phylogenetic relationship among human- and dog-associated UPEC? Can the author describe this point in the introduction?

Lines 49-51 Can the authors explain the mechanism through which the biofilm is associated to recurrent infections?

Lines 52-54 To be precise, IBCs are bacterial communities with biofilm-like properties. Moreover, the capability to form IBCs depends mainly on the ability of bacteria to be internalized within a eukaryotic cell rather than their biofilm forming activity. Please, rephrase the paragraph.

Lines 63-64 What did the authors mean for antimicrobials? Also molecules from innate immunity?

Lines 71-73 See comment above about the genetic/phylogenetic relationship among human- and dog-associated UPEC?

Lines 76-77 What did the authors mean for “similar uropathogens”?

Line 79 Please, replace “cytologies” with “cytology” throughout the text.

Line 108 Did the authors confirm the phenotypic resistance to beta-lactams?

Line 277 Ref 23 is not correct. I suggest to replace it with the following: 10.3390/molecules25020316; 10.1186/s12879-018-3467-0.

Line 335 In my opinion letters are confusing. Please, report statistically significant differences by using asterisks.

Lines 252-253 This sentence is not correct. There are several papers describing the characteristics of E. coli responsible for recurrent UTIs in humans. Probably the authors refer to those associated with dogs.

Major points:

Lines 123-125 Did the authors perform real time PCR or PCR? Evaluation of gene expression seems in contrast to the results described in lines 107-108. Can the authors clarify this point? What did the author mean for “standard screening with the Clinical Microbiology Laboratory for any Escherichia coli”

Line 145 Fig. 3 shows huge bacterial communities located beneath the superficial layers of the bladder. Is this right? Usually IBCs are observed within cells at the superficial layer of the epithelium. Can the authors describe this point? I suggest also to insert arrows showing IBCs.

Line 180 Bars in figure 4 should show the IBC-CFU/ml counting for each tested strains to point out differences among strains. In this form the bars are not understandable. Moreover, negative and positive controls should be added in these type of analyses. A lab E. coli strain can be used as negative control. Vice versa, an already reported IBC producing strain should be used as positive control.

Line 229 Did these fimH negative strains belong to the 30 analyzed? In line 180 the authors reported 5 strains with no capability to form IBCs. Can the authors clarify the origin of these strains? How did the authors evaluate the absence of fimH gene?

Lines 260-261 Is this analysis statistically valid? Considering the low number of analyzed isolates anything can be concluded.

Lines 277-278 FIG S2B is not mentioned in the text and due to the lack of significant results I would suggest the authors to delete totally this analysis.

Lines 310-317 These results are not very solid. Are the antibiotics tested capable to permeate epithelial cells? If not, no conclusions can be drawn from these experiments. I suggest to use an antibiotic that penetrate the cells to evaluate if bacteria forming IBCs are somehow resistant or not.

Major points:

Discussion section should be shortened

Lines 338-340 Please, lower your enthusiasms about the implications of your model. This type of conclusion cannot be drawn based on the reported results.

Lines 348-349 As stated before, this conclusion cannot be drawn.

Lines 350-351 As stated before, this conclusion cannot be drawn.

Line 442 See comment above. Are these antibiotics able to permeate the bladder cells?

Line 462 With the current knowledge it is very difficult to assign one specific pathotype to an E. coli isolate. So it would be better to increase the number of gene tested to increase the probability of a correct classifications. The list of genes associated with UPEC isolates is reported in https://doi.org/10.1007/s10123-022-00235-y and references therein.

Minor points:

Line 335 Please, use only the acronym.

Line 357 Can the authors compare these rates with published data?

Lines 408-410 Are there evidences of the presence of other adhesins involved in bacterial internalization?

Lines 433-434 In my opinion, a model for canine related UTIs is needed to study canine UTIs. Now there are several advanced models based on human cells to evaluate human UTIs.

Line 437 What is the meaning of one health model? Probably it would be better to highlight the fact that UPEC strains can infect humans and dogs, if this is correct. For example, are there studies describing the relationship among human and canine isolates?

Reviewer #10: About the article, I think the authors replied to all comments; therefore, the paper is acceptable, now.

Sincerely

Reviewer #11: In this manuscript, Gilbertie et al provide a characterization of the role of intracellular bacterial communities (IBCs) in urinary tract infections in dogs. The authors have addressed several of the concerns of the reviewers from Round 1, but overall, there are still areas where the manuscript can be improved, and the claims clarified. The review was also made very difficult by the fact that there are many versions of the text, including one with comments between the authors still included. It was therefore hard to access which is the final revised text – the comments below are based on the text with the track changes on. Figure 2 is also missing from the revised manuscript, and Figure S2 is repeated as ‘Figure 2 revised’ which is incorrect.

Major comments:

IBCs vs intracellular bacteria: It is unclear to me how the authors define and identify IBCs. At the first glance, it appears that the manuscript appears to characterize all intracellular growth of UPEC as IBCs, especially in the later figures. IBCs are only a subset of intracellular bacteria and are characterized by a high intracellular load of bacteria in umbrella cells, with many of the bacteria within the IBCs being smaller and of coccoid shape. These sorts of structures are evident in Figure 3, where the authors clearly indicate these structures in Figure 3 so that the reader can appreciate them better. Thereafter, some clarity is needed as to how IBCs were enumerated in the high-power field images, and how the total bacterial load was enumerated. If the authors have treated every instance of intracellular growth as an IBC, then they should use the term intracellular bacteria rather than IBCs, especially for the data in Figures 5 and 6. If not, then pointing out examples of IBCs, especially in the FimH+ samples would help the reader.

Antibiotic treatment data: Please show the individual points in the bar plots of Figure 7 (as shown in the other figures) to help the reader appreciate the spread in CFU. What do the letters a, b represent in the statistical comparisons – the Figure legend does not clarify this either. Please provide p values from an ANOVA test corrected for multiple comparisons. There does appear to be quite a significant decrease in the number of IBCs in the treatment with Enro. The authors need to discuss the differences between antibiotics more clearly in the Main Text and Discussion. It is also unclear why an arbitrary 2log threshold is cited in the Main Text as the cutoff for any significant difference in CFU. The duration of antibiotic treatment is also unclear. These methodological points need to be made clear and if a comparison is made to the results in the bladder-on-chip study that the authors reference in the Discussion, then it is important to note that there was some clearance by Ampicillin in that study (but very delayed compared to clearance of lumenal bacteria.

Minor points

Recurrent UTIs: the main text should be edited to clarify that these are canine-specific strains.

Figure 1: Percentage numbers are missing on the figure-1B and 1C.

Amalgamation of some figures: The manuscript would be improved if the number of main figures were reduced. For example, the low power field images in Figure 4 could be moved to a supplementary, and Figures 5 and 6 could be merged to help the reader identify the clear trend FimH-< FimH+<fimh+ recurrent.="">

Use of colours in the figures in the paper: The authors could use non-primary colours to enable colour-blind readers to appreciate the images better. UPEC are identified as labelled green in Figures 3 and 4 when they are in fact shown in yellow.

Role of other tissue-resident populations in persistent infections: Quiescent intracellular bacteria in the bladder PMID: 16968784 are also thought to contribute to recurrence; similar populations of solitary bacteria were also observed in a recent study with mouse bladder organoids PMID: 34289360. Could the authors comment on whether similar mechanisms might operate in the canine bladder and whether some of the bacteria they observe in the canine tissue might belong to these sub-populations?

Additional papers on human IBCs in human epithelial cells: At Line 171 or 181 depending on whether track changes are off or on, the authors could cite these relevant papers with UTI work in human cell lines: PMID: 32540870, 29352171 in addition to the study with the PD07i line.

Comparison with murine models for UTIs in the Discussion: At line 389 or 414 depending on whether track changes is off or on, the authors should also include murine bladder organoids PMID: 34289360 in the discussion, where IBC formation was also shown and visualized by confocal and volumetric electron microscopy.

Comparison with human models in the Discussion: At lines 390 or 415, the authors should also include references to Transwell models that have been used by several groups PMID: 29352171, 37051302</fimh+>

7. PLOS authors have the option to publish the peer review history of their article (what does this mean?). If published, this will include your full peer review and any attached files.

Reviewer #1: No

Reviewer #3: **Yes: **Payam BEHZADI

Reviewer #4: **Yes: **Paola Scavone

Reviewer #5: No

Reviewer #6: No

Reviewer #7: No

Reviewer #8: No

Reviewer #9: No

Reviewer #10: **Yes: **Seyedeh Elham Rezatofighi

Reviewer #11: No

---

## [Author Response · Author response to Decision Letter 1]

14 Nov 2023

We thank the reviewers for their continued support and investment in this manuscript. With 11 reviewers, it is challenging to address all comments, but we have provided a point by point response below and continue to look for ways to improve our work. 

Reviewer #3

Thank you for your manuscript submission. I believe that your study has fundamental problem. As you know, E.coli has a powerful arsenal including a wide range of virulence factors. You can not draw a conclusion on the basis of a virulence factor like FimH or other virulence factor in solo regarding a feature like IBC. There are many paradox in your manuscript which each part deny the other. Hence, my decision regarding the present manuscript is Definite Rejection.

We appreciate your opinion; we never state that FimH is the only gene responsible for IBC formation. We evaluated this gene as it has been shown in other publications in other species to play a vital role in the first step of IBC formation. Therefore, we examined if this gene was important in canine IBC, and we found that it was. As this is in accordance with studies in other mammalian species, it serves to support that IBC pathophysiology is similar in dogs which is the basis of this comparative model. In future studies, we plan to examine additional virulence factors. The goal of this paper, however, is to establish the dog as a relevant UTI model, and additional virulence determination is beyond the scope of this paper.

Reviewer #4

The only concern that has no been addressed is related with the microscope technique selected for visualization and quantification of IBC. Epifluorescence do not allow to differentiate if a bacteria is on the surface of the cell or inside because of the type of images it gave. In this cases, confocal microscopy is the best option as it is possible to take optical images in the z-plane. Confocal microscopy will confirm that the bacteria are really inside the cell and not in the surface.

Thank you for your concern. We believe we have addressed this possibility in the methods section where we describe using gentamicin to kill any extracellular bacteria starting 1 hr post infection. The gentamicin remains in the media until the end of the experiment. We also wash extensively before imaging to remove dead organisms. 

Reviewer #6 

Please ensure the manuscript is submitted as per guidelines of the journal. This is interesting study, and hard work has been done to provide ne information, but there is significantly change needed to write in research manuscript format. Specific comments are provided in the manuscript. The author has made change as requested from previous review. However, the manuscript is not up to the standard of journal. The method section has all previous work description, and some current work description. Result section has method and discussion.

Thank you for your notes on the formatting of the manuscript. We have simplified several of the results section in this version to remove more of the methods. However, we feel that having a brief leading description of the experiment leads to a better understanding of the results we are presenting. We have formatted our sections like this in other manuscripts submitted to PLoS with success. We hope that the edits we have made have addressed this comment appropriately.

The ANOVA analysis performed, and did not mention the result. For what section did you performed ANOVA? Please elaborate. The figure still needs clear caption. Results for each experiment conducted in this study would added value in terms of data sharing as there are only 9 sample size. 

In the statistical analysis, we state the ANOVA was used to assess the antimicrobial treatments. In that specific figure legend (Fig 7), we state that an ANOVA was used. The results of the ANOVA are presented in the figure as differing letters indicating p<0.05 based on the Tukey post hoc analysis. However, we have removed the letters and instead our now presenting the p-values of the untreated control (NT) against each individual antimicrobial treatment (ENRO, POD, DOX). We hope that this presentation is what the reviewer was looking for.

Figure with IBCs and bar graph do not provide enough data to researcher who want to replicate the study in future.

We understand that the bar graphs may not give enough detail for replication. Therefore, we changed the bar graphs to a bar graph with an overlaid scatter plot so individuals can see each individual data point. We hope that aids in the replicability of the data.

Reviewer #7

Authors have written in the Ethics statement that IACUC approval was not required. I think this statement should be revisited.

We have addressed the ethics statement and IACUC need in the manuscript with justification. If additional information is needed, please be specific what is missing so we can be more clear.

Reviewer #8

This study is aimed to investigate whether dogs evidence similar UPEC-associated UTI pathogenesis with IBC formation, and to validate an in vitro IBC model using primary canine urothelial cells. The work overall is solid, although I do have some comments (noted below).

Comments

- In all figures, font types are unreadable, please modify. Similar problems arise with the scale bar in most of the micrographs. We have edited the figures to increase legibility. 

- Information regarding the retrospective study performed should be added in Materials and Methods Section, including its ethical approval. We have added information regarding the retrospective analysis into the methods

- L334. Data regarding antimicrobial susceptibility should be shown. We have added the MIC breakpoints to line 334 in replace of “data not shown”.

- Authors should proofread the manuscript and correct some minor spelling mistakes. The manuscript has been re-proofread to catch any spelling mistakes.

Review #9

The authors described the capability of uropathogenic Escherichia coli strains, isolated from canine UTIs, to invade bladder cells and to form the intracellular bacterial communities (IBCs). This represents a very interesting topic for microbiologists characterizing UPEC strains and the paper provides a basis to study the genotypic and phenotypic relationship between human and dog associated UPEC. Images of IBCs are solid and results reported in the manuscript are quite in line with the published data. However, authors should address the following points:

Minor points:

Introduction section should be improved. We have updated this section, however, this comment is not particularly specific so if holes are noted please let us know.

Lines 44-47 Is there any genetic/phylogenetic relationship among human- and dog-associated UPEC? Can the author describe this point in the introduction? The only comparisons that have been performed are presented in paragraph 4. This work is currently being performed in the first author’s laboratory as a collaboration between a veterinary school and medical school in the same geographic region. We expect that manuscript to follow this one in the next year.

Lines 49-51 Can the authors explain the mechanism through which the biofilm is associated to recurrent infections? In paragraph 2 we state that had no found a correlation between traditional surface attached biofilms and strains that cause recurrent infections. Therefore, IBCs, which are biofilm-like, may be associated but yet proven.

Lines 52-54 To be precise, IBCs are bacterial communities with biofilm-like properties. Moreover, the capability to form IBCs depends mainly on the ability of bacteria to be internalized within a eukaryotic cell rather than their biofilm forming activity. Please, rephrase the paragraph. This paragraph has been rephrased.

Lines 63-64 What did the authors mean for antimicrobials? Also molecules from innate immunity? This has been edited for clarity to mean prescribed antimicrobial agents not naturally derived from the innate immune response.

Lines 71-73 See comment above about the genetic/phylogenetic relationship among human- and dog-associated UPEC? To date, there is nothing in the literature and this manuscript focuses on IBC pathophysiology. However, this work is currently the focus of a grant in the first authors laboratory.

Lines 76-77 What did the authors mean for “similar uropathogens”? This has been changed to etiologic agents.

Line 79 Please, replace “cytologies” with “cytology” throughout the text. As we are referring to multiple not individual we would argue to keep as cytologies, similar to biopsy vs biopsies. 

Line 108 Did the authors confirm the phenotypic resistance to beta-lactams? Yes, this has been rephrased to indicate this.

Line 277 Ref 23 is not correct. I suggest to replace it with the following: 10.3390/molecules25020316; 10.1186/s12879-018-3467-0. This has been corrected.

Line 335 In my opinion letters are confusing. Please, report statistically significant differences by using asterisks. This has been changed to astericks.

Lines 252-253 This sentence is not correct. There are several papers describing the characteristics of E. coli responsible for recurrent UTIs in humans. Probably the authors refer to those associated with dogs. The authors could only find references showing that recurrent strains carry more virulence genes but no specific genes have been associated apart from fimH which we evaluate in this manuscript. In addition, there is controversy over the associated between surface-associated biofilm grown in a microtiter plate correlating with recurrent infections. Which is why the authors propose that IBCs may have a higher association. Nevertheless, we have rephrased this to better explain our literature findings.

Major points:

Lines 123-125 Did the authors perform real time PCR or PCR? Evaluation of gene expression seems in contrast to the results described in lines 107-108. Can the authors clarify this point? What did the author mean for “standard screening with the Clinical Microbiology Laboratory for any Escherichia coli”

The clinical assessment of virulence and antimicrobial susceptibility as was done when collecting these isolates through clinical canine submissions was performed by conventional PCR. We have corrected the description that inferred real time PCR. 

Line 145 Fig. 3 shows huge bacterial communities located beneath the superficial layers of the bladder. Is this right? Usually IBCs are observed within cells at the superficial layer of the epithelium. Can the authors describe this point? I suggest also to insert arrows showing IBCs.

The authors note that they did not point out the autofluorescence that hemorrhage within the bladder epithelium causes. Therefore, we have added a note about that in the figure legend and added arrows to denote the IBCs in each image.

Line 180 Bars in figure 4 should show the IBC-CFU/ml counting for each tested strains to point out differences among strains. In this form the bars are not understandable. Moreover, negative and positive controls should be added in these type of analyses. A lab E. coli strain can be used as negative control. Vice versa, an already reported IBC producing strain should be used as positive control.

We have changed the bars to scatter plots with bars to aid here. As UTI89 is not a canine strain nor any other published strains, we do not have a positive and negative controls specific for our species. We have robust IBC formation noted by several of our strains in addition to those that form no IBCs at all. Therefore, our screening has identified strains we can use moving forward as positive and negative canine-specific controls.

Line 229 Did these fimH negative strains belong to the 30 analyzed? In line 180 the authors reported 5 strains with no capability to form IBCs. Can the authors clarify the origin of these strains? How did the authors evaluate the absence of fimH gene?

Only 3 of the fimH negative strains belonged to the 30 analyzed. We used our repository of 206 to find the other 6. We only had 9 total in all 206. We added this to the text to further clarify.

Lines 260-261 Is this analysis statistically valid? Considering the low number of analyzed isolates anything can be concluded.

Yes, this is statistically valid. Now that the scatter plots are shown you can see how distinct the populations are further supporting the statistics.

Lines 277-278 FIG S2B is not mentioned in the text and due to the lack of significant results I would suggest the authors to delete totally this analysis.

We have removed this graph from the supplemental figure.

Lines 310-317 These results are not very solid. Are the antibiotics tested capable to permeate epithelial cells? If not, no conclusions can be drawn from these experiments. I suggest to use an antibiotic that penetrate the cells to evaluate if bacteria forming IBCs are somehow resistant or not.

We have added text to further clarify this. Even though cephalosporins have lower intracellular penetration then fluoroquinolones, we used their known penetration rates to ensure that the MIC was achieved within the cell. However, we understand that this is not well described and has now been updated throughout the text.

Major points:

Discussion section should be shortened We have removed a paragraph.

Lines 338-340 Please, lower your enthusiasms about the implications of your model. This type of conclusion cannot be drawn based on the reported results. We changed the use of robust to potential to dampen the enthusiasm.

Lines 348-349 As stated before, this conclusion cannot be drawn. See above.

Lines 350-351 As stated before, this conclusion cannot be drawn. See above.

Line 442 See comment above. Are these antibiotics able to permeate the bladder cells? See above.

Line 462 With the current knowledge it is very difficult to assign one specific pathotype to an E. coli isolate. So it would be better to increase the number of gene tested to increase the probability of a correct classifications. The list of genes associated with UPEC isolates is reported in https://doi.org/10.1007/s10123-022-00235-y and references therein.

We appreciate this evolving field and the additional characterization that is available. At this point, we are most interested in replicating intracellular bacterial communities with canine UTI strains; we could continue to further characterize these isolates but for use in this purpose of this study we don’t think it would change or enhance our findings. 

Minor points:

Line 335 Please, use only the acronym. This has been corrected

Line 357 Can the authors compare these rates with published data? We have added this.

Lines 408-410 Are there evidences of the presence of other adhesins involved in bacterial internalization? Not in bladder epithelium. In kidney cells, the P fimbriae are important. In the intestine a variety depending on the pathotype and location are important.

Lines 433-434 In my opinion, a model for canine related UTIs is needed to study canine UTIs. Now there are several advanced models based on human cells to evaluate human UTIs. We disagree respectfully. In our opinion as translational scientists, having large animal models of spontaneous disease can fit the NIH’s species definition of “Dual Purpose with Dual Benefit”. In that discoveries made in the canine model can be translated to human medicine. If you think about new drugs that need to be tested in relevant models, the canine model could be the key step between murine experiments and first-in-human trials. In fact, clinical trials performed in dogs for relevant diseases can be used as preclinical data supporting IND submissions to the FDA. Therefore, showing that the pathophysiology for UPEC-associated UTIs is the first step towards this.

Line 437 What is the meaning of one health model? Probably it would be better to highlight the fact that UPEC strains can infect humans and dogs, if this is correct. For example, are there studies describing the relationship among human and canine isolates? As stated in the above response, One Health approaches are powerful tools to advance both veterinary and human medicine. If one can show similarities in disease processes, than findings i

---

## [Decision Letter · Decision Letter 2]

5 Dec 2023

PONE-D-22-30262R2Canine urothelial cell model to study intracellular bacterial community development by uropathogenic Escherichia coliPLOS ONE

Dear Dr. Jacob,

Thank you for submitting your manuscript to PLOS ONE. After careful consideration, we feel that it has merit but does not fully meet PLOS ONE’s publication criteria as it currently stands. Therefore, we invite you to submit a revised version of the manuscript that addresses the points raised during the review process.

We look forward to receiving your revised manuscript.

Kind regards,

Kwame Kumi Asare, Ph.D

Academic Editor

PLOS ONE

Journal Requirements:

Reviewers' comments:

Reviewer's Responses to Questions

**Comments to the Author**

1. If the authors have adequately addressed your comments raised in a previous round of review and you feel that this manuscript is now acceptable for publication, you may indicate that here to bypass the “Comments to the Author” section, enter your conflict of interest statement in the “Confidential to Editor” section, and submit your "Accept" recommendation.

Reviewer #1: All comments have been addressed

Reviewer #3: (No Response)

Reviewer #6: (No Response)

Reviewer #8: All comments have been addressed

Reviewer #9: All comments have been addressed

Reviewer #10: All comments have been addressed

Reviewer #11: (No Response)

2. Is the manuscript technically sound, and do the data support the conclusions?

Reviewer #1: (No Response)

Reviewer #3: (No Response)

Reviewer #6: Partly

Reviewer #8: Yes

Reviewer #9: Yes

Reviewer #10: Yes

Reviewer #11: Yes

3. Has the statistical analysis been performed appropriately and rigorously? 

Reviewer #1: (No Response)

Reviewer #3: (No Response)

Reviewer #6: (No Response)

Reviewer #8: Yes

Reviewer #9: Yes

Reviewer #10: Yes

Reviewer #11: Yes

4. Have the authors made all data underlying the findings in their manuscript fully available?

Reviewer #1: (No Response)

Reviewer #3: (No Response)

Reviewer #6: Yes

Reviewer #8: Yes

Reviewer #9: Yes

Reviewer #10: Yes

Reviewer #11: Yes

5. Is the manuscript presented in an intelligible fashion and written in standard English?

Reviewer #1: (No Response)

Reviewer #3: (No Response)

Reviewer #6: No

Reviewer #8: Yes

Reviewer #9: Yes

Reviewer #10: Yes

Reviewer #11: No

6. Review Comments to the Author

Reviewer #1: a;lskdfja;lksdfjlaskjflkasjfl;asjfdla;sdfkjla;sdkjf;lasdkfjals;dfkjas ldfja;lsdkfj ;sd;fkasd'f;lkasdl;fkas'df;l

Reviewer #3: (No Response)

Reviewer #6: (No Response)

Reviewer #8: This is a revised version. In this improved manuscript, the authors clearly answered the reviewer´s comments.

Reviewer #9: MINOR REVISION

Lines 56-57 of the revised manuscript track change on, Several papers described the association between IBCs and rUTIs in humans. The authors should underline that this association is still unknown for canine rUTIs.

Please, use E. coli instead of Escherichia coli. Correct it throughout the text.

Bars in Fig 4 are still not understandable and informative. The authors should underline that the tested isolates showed high variability in IBCs formation capacity. There are isolates with no capability to be internalized within bladder cells, vice versa there are isolates forming IBCs. Moreover, CFU/ml counting should be expressed for each isolates known to form IBCs. Finally, fig legend has some mistakes, please revise it.

Lines 201-203 of the revised manuscript track change on. The size cannot be expressed in molarity.

Line 536 of the revised manuscript track change on, the ref 26 does not report the PCR method used to verify the presence of ExPEC associated genes. Which primers did the authors use? Can the authors add the more appropriate reference?

Reviewer #10: Dear Authors and Editor

The manuscript quality and clarity improved from the last version. The authors addressed all the points I made. Therefore, I think it can be accepted.

Sincerely

Reviewer #11: The authors have addressed all major points raised in the previous round of review. They are to be commended for their efforts in addressing the responses of so many reviewers. In particular, the larger images with better labelling are visually more striking, and the use of arrows makes it straightforward to identify features of interest. They have also clearly demonstrated that they are identifying IBCs and not intracellular bacteria in general within the tissues. The new text on the effects of antibiotic treatment also clarifies better the nature of the different antibiotics used and how this reflects in the results that the authors observe. We agree with the authors that the citation list is already long, and so leave it to their judgement whether to take on board the suggestions we made in the last round of review.

There are only some minor points that need editing before acceptance:

1. The exact duration of antibiotic treatment is still not mentioned. Could the authors include this in the Main Text?

2. It would be helpful to the reader if the correspondence between the number of stars and the p values are also mentioned in the Figure Legends wherever appropriate.

3. Line 54: “To date, it is unknown if IBCs play a role in UTI recurrence” some of the evidences regarding IBC playing a role in UTI recurrence can be recognized from movies from mouse models by Justice et. al (2004) and bladder-on-a-chip model by Sharma et.al (2021). Consider rephrasing the sentence.

4. Line 176 – 178 and Figure S2a: the unit should be micron and not micromolar. Please use a small ‘m’.

5. Line 261: Figure S3 not S2.

6. The figure S3 is not referenced anywhere in the text. Please also mention the p values for this figure.

7. Line 292: ‘with’ appears twice in the sentence.

8. Line 293: typo µg/mL

9. Line 297: what exactly do the authors mean by ‘eradicate considerably’? Do they mean the complete absence of IBCs. Please edit for clarity.

10. Line 306: could the authors add the appropriate reference for the C/E values for the antibiotics as this is the first time that this parameter is mentioned?

11. Line 313. Delete one of the extra periods at the end of the sentence.

12. Figure 7B. The X axis labels are missing, and the description of how the CFU values are obtained is unclear based on the text given in the legend. Are the authors measuring the total bacterial load per well (intracellular and extracellular)? What does ‘unfixed to release the release the intracellular bacteria’ mean? Does it mean that the host cells were lysed?

13. Reference duplication: Reference 41 is the same as reference 39, and reference 65 is the same as reference 46. Did the authors mean to refer to other papers written by the same authors in each case?

7. PLOS authors have the option to publish the peer review history of their article (what does this mean?). If published, this will include your full peer review and any attached files.

Reviewer #1: No

Reviewer #3: **Yes: **Payam BEHZADI

Reviewer #6: No

Reviewer #8: No

Reviewer #9: No

Reviewer #10: **Yes: **Seyedeh Elham Rezatofighi

Reviewer #11: No

---

## [Author Response · Author response to Decision Letter 2]

12 Sep 2024

Review #9

Lines 56-57 of the revised manuscript track change on, Several papers described the association between IBCs and rUTIs in humans. The authors should underline that this association is still unknown for canine rUTIs. Corrected.

Please, use E. coli instead of Escherichia coli. Correct it throughout the text. Corrected.

Bars in Fig 4 are still not understandable and informative. The authors should underline that the tested isolates showed high variability in IBCs formation capacity. There are isolates with no capability to be internalized within bladder cells, vice versa there are isolates forming IBCs. Moreover, CFU/ml counting should be expressed for each isolates known to form IBCs. Finally, fig legend has some mistakes, please revise it. Changed from bars to lines. The black lines denote the mean and standard deviation and the open blue circles represent the individual UPEC strains. The figure legend has also be updated and edited.

Lines 201-203 of the revised manuscript track change on. The size cannot be expressed in molarity. Corrected.

Line 536 of the revised manuscript track change on, the ref 26 does not report the PCR method used to verify the presence of ExPEC associated genes. Which primers did the authors use? Can the authors add the more appropriate reference? Corrected.

Reviewer 11

1. The exact duration of antibiotic treatment is still not mentioned. Could the authors include this in the Main Text? Corrected.

2. It would be helpful to the reader if the correspondence between the number of stars and the p values are also mentioned in the Figure Legends wherever appropriate. Corrected.

3. Line 54: “To date, it is unknown if IBCs play a role in UTI recurrence” some of the evidences regarding IBC playing a role in UTI recurrence can be recognized from movies from mouse models by Justice et. al (2004) and bladder-on-a-chip model by Sharma et.al (2021). Consider rephrasing the sentence. Corrected.

4. Line 176 – 178 and Figure S2a: the unit should be micron and not micromolar. Please use a small ‘m’. Corrected.

5. Line 261: Figure S3 not S2. Corrected.

6. The figure S3 is not referenced anywhere in the text. Please also mention the p values for this figure. Corrected.

7. Line 292: ‘with’ appears twice in the sentence. Corrected.

8. Line 293: typo µg/mL Corrected.

9. Line 297: what exactly do the authors mean by ‘eradicate considerably’? Do they mean the complete absence of IBCs. Please edit for clarity. Amended for clarity with numbers in parentheses.

10. Line 306: could the authors add the appropriate reference for the C/E values for the antibiotics as this is the first time that this parameter is mentioned? We are unclear what C/E means. We are reporting Cmax or peak concentration in urine based on the referenced canine pharmacokinetics studies. We attempted to reword the sentence to clarify this statement better.

11. Line 313. Delete one of the extra periods at the end of the sentence. Corrected.

12. Figure 7B. The X axis labels are missing, and the description of how the CFU values are obtained is unclear based on the text given in the legend. Are the authors measuring the total bacterial load per well (intracellular and extracellular)? What does ‘unfixed to release the release the intracellular bacteria’ mean? Does it mean that the host cells were lysed? Corrected for clarity. The gentamicin kills extracellular bacteria, so when we lyse the urothelial cells we are only quantifying the intracellular bacteria.

13. Reference duplication: Reference 41 is the same as reference 39, and reference 65 is the same as reference 46. Did the authors mean to refer to other papers written by the same authors in each case? Reference manager issues that have now been corrected.

---

## [Decision Letter · Decision Letter 3]

30 Oct 2024

PONE-D-22-30262R3Canine urothelial cell model to study intracellular bacterial community development by uropathogenic Escherichia coliPLOS ONE

Dear Dr. Jacob,

Thank you for submitting your manuscript to PLOS ONE. After careful consideration, we feel that it has merit but does not fully meet PLOS ONE’s publication criteria as it currently stands. Therefore, we invite you to submit a revised version of the manuscript that addresses the points raised during the review process.

We look forward to receiving your revised manuscript.

Kind regards,

Kwame Kumi Asare, Ph.D

Academic Editor

PLOS ONE

Reviewers' comments:

Reviewer's Responses to Questions

**Comments to the Author**

1. If the authors have adequately addressed your comments raised in a previous round of review and you feel that this manuscript is now acceptable for publication, you may indicate that here to bypass the “Comments to the Author” section, enter your conflict of interest statement in the “Confidential to Editor” section, and submit your "Accept" recommendation.

Reviewer #9: (No Response)

Reviewer #11: (No Response)

2. Is the manuscript technically sound, and do the data support the conclusions?

Reviewer #9: Partly

Reviewer #11: Yes

3. Has the statistical analysis been performed appropriately and rigorously? 

Reviewer #9: Yes

Reviewer #11: Yes

4. Have the authors made all data underlying the findings in their manuscript fully available?

Reviewer #9: Yes

Reviewer #11: (No Response)

5. Is the manuscript presented in an intelligible fashion and written in standard English?

Reviewer #9: Yes

Reviewer #11: Yes

6. Review Comments to the Author

Reviewer #9: The authors addressed some of the point raised at the first round of revision. However, the following point lacks of response.

"Lines 310-317 These results are not very solid. Are the antibiotics tested capable to permeate epithelial cells? If not, no conclusions can be drawn from these experiments. I suggest to use an antibiotic that penetrate the cells to evaluate if bacteria forming IBCs are somehow resistant or not."

Reviewer #11: All issues raised in the previous round have been satisfactority addressed, except the one about the duplicates in the reference list. In version R3, References 39 and 88 are the same, as are 50 and 69. Please check these to ensure that they are not meant to refer to different papers by people with the same name prior to acceptance.

7. PLOS authors have the option to publish the peer review history of their article (what does this mean?). If published, this will include your full peer review and any attached files.

Reviewer #9: No

Reviewer #11: No

---

## [Author Response · Author response to Decision Letter 3]

6 Dec 2024

Reviewer #9: The authors addressed some of the point raised at the first round of revision. However, the following point lacks of response.

"Lines 310-317 These results are not very solid. Are the antibiotics tested capable to permeate epithelial cells? If not, no conclusions can be drawn from these experiments. I suggest to use an antibiotic that penetrate the cells to evaluate if bacteria forming IBCs are somehow resistant or not."

Yes, we did use and have added two references on the two antimicrobials with intracellular activity (doxycycline and enrofloxacin; Line 363). 

Reviewer #11: All issues raised in the previous round have been satisfactority addressed, except the one about the duplicates in the reference list. In version R3, References 39 and 88 are the same, as are 50 and 69. Please check these to ensure that they are not meant to refer to different papers by people with the same name prior to acceptance.

We appreciate you pointing this out. We have checked the references and removed the those duplicates.

---

## [Decision Letter · Decision Letter 4]

18 Dec 2024

Canine urothelial cell model to study intracellular bacterial community development by uropathogenic Escherichia coli

PONE-D-22-30262R4

Dear Dr. Jacob,

We’re pleased to inform you that your manuscript has been judged scientifically suitable for publication and will be formally accepted for publication once it meets all outstanding technical requirements.

Kind regards,

Kwame Kumi Asare, Ph.D

Academic Editor

PLOS ONE

Additional Editor Comments (optional):

Reviewers' comments:

Reviewer's Responses to Questions

**Comments to the Author**

1. If the authors have adequately addressed your comments raised in a previous round of review and you feel that this manuscript is now acceptable for publication, you may indicate that here to bypass the “Comments to the Author” section, enter your conflict of interest statement in the “Confidential to Editor” section, and submit your "Accept" recommendation.

Reviewer #9: All comments have been addressed

Reviewer #11: All comments have been addressed

2. Is the manuscript technically sound, and do the data support the conclusions?

Reviewer #9: Yes

Reviewer #11: Yes

3. Has the statistical analysis been performed appropriately and rigorously? 

Reviewer #9: Yes

Reviewer #11: Yes

4. Have the authors made all data underlying the findings in their manuscript fully available?

Reviewer #9: Yes

Reviewer #11: Yes

5. Is the manuscript presented in an intelligible fashion and written in standard English?

Reviewer #9: Yes

Reviewer #11: Yes

6. Review Comments to the Author

Reviewer #9: (No Response)

Reviewer #11: (No Response)

7. PLOS authors have the option to publish the peer review history of their article (what does this mean?). If published, this will include your full peer review and any attached files.

Reviewer #9: No

Reviewer #11: No

---

## [Editor Report · Acceptance letter]

26 Dec 2024

PONE-D-22-30262R4 

PLOS ONE

Dear Dr. Jacob, 

I'm pleased to inform you that your manuscript has been deemed suitable for publication in PLOS ONE. Congratulations! Your manuscript is now being handed over to our production team.

Kind regards, 

on behalf of

Dr. Kwame Kumi Asare 

Academic Editor

PLOS ONE